



# IWV observations in the Caribbean Arc from a network of ground-based GNSS receivers during EUREC[4]A

Olivier Bock[1,2], Pierre Bosser[3], Cyrille Flamant[4], Erik Doerflinger[5], Friedhelm Jansen[6], Romain Fages[7], Sandrine Bony[8], and Sabrina Schnitt[9]

[1]Université de Paris, Institut de physique du globe de Paris, CNRS, IGN, F-75005 Paris, France
[2]ENSG-Géomatique, IGN, F-77455 Marne-la-Vallée, France
[3]Lab-STICC UMR 6285 CNRS / PRASYS, ENSTA Bretagne / HOP, F-29200 Brest, France
[4]LATMOS/IPSL, UMR 8190 CNRS-SU-UVSQ, Paris, France
[5]Géosiences Montpellier UMR 5243, Université Montpellier - CNRS, Montpellier, France
[6]Max-Planck-Institut für Meteorologie, Hamburg, Germany
[7]IGN, Saint-Mandé, France
[8]LMD/IPSL, UMR 8539 CNRS, Sorbonne Université, Paris, France
[9]Institute for Geophysics and Meteorology, University of Cologne, Cologne, Germany

**Correspondence:** Olivier Bock (bock@ipgp.fr)

**Abstract.** Ground-based Global Navigation Satellite System (GNSS) measurements from nearly fifty stations distributed over the Caribbean Arc have been analysed for the period 1 January-29 February 2020 in the framework of the EUREC[4]A (Elucidate the Couplings Between Clouds, Convection and Circulation) field campaign. The aim of this effort is to deliver high-quality Integrated Water Vapour (IWV) estimates to investigate the moisture environment of mesoscale cloud patterns in the Tradewinds

and their feedback on the large-scale circulation and energy budget.

This paper describes the GNSS data processing procedures and assesses the quality of the GNSS IWV retrievals from four operational streams and one reprocessed research stream which is the main data set used for offline scientific applications. The uncertainties associated with each of the data sets, including the zenith tropospheric delay (ZTD) to IWV conversion methods and auxiliary data, are quantified and discussed. The IWV estimates from the reprocessed data set are compared to

the Vaisala RS41 radiosonde measurements operated from the Barbados Cloud Observatory (BCO) and to the measurements from the operational radiosonde station at Grantley Adams international airport (GAIA). A significant dry bias is found in the GAIA humidity observations with respect to the BCO sondes (-2.9 $\mathrm{kg\,m^{-2}}$) and the GNSS results (-1.2 $\mathrm{kg\,m^{-2}}$). A systematic bias between the BCO sondes and GNSS is also observed (1.7 $\mathrm{kg\,m^{-2}}$) where the Vaisala RS41 measurements are moister than the GNSS retrievals. The IWV estimates from a colocated microwave radiometer agree with the BCO soundings after an

instrumental update on 27 January, while they exhibit a dry bias compared to the soundings and to GNSS before that date. IWV estimates from the ECMWF fifth generation reanalysis (ERA5) are overall close to the GAIA observations, probably due to the assimilation of these observations in the reanalysis. However, during several events where strong peaks in IWV occurred, ERA5 is shown to significantly underestimate the GNSS derived IWV peaks. Two successive peaks are observed on 22 January and 23/24 January which were associated with heavy rain and deep moist layers extending from the surface up





to altitudes of 3.5 and 5 km, respectively. ERA5 significantly underestimates the moisture content in the upper part of these layers. The origins of the various moisture biases are currently being investigated.

We classified the cloud organisation for five representative GNSS stations across the Caribbean Arc and found that the environment of Fish cloud patterns to be moister than that of Flowers cloud patterns which, in turn, is moister than the environment of Gravel cloud patterns. The differences in the IWV means between Fish and Gravel were assessed to be significant. Finally,

the Gravel moisture environment was found to be similar to that of clear, cloud-free conditions. The moisture environment associated with the Sugar cloud pattern has not been assessed because it was hardly observed during the first two months of 2020.

The reprocessed ZTD and IWV data set from 49 GNSS stations used in this study are available from the AERIS data center (https://doi.org/10.25326/79; Bock (2020b)).

# 1 Introduction

The overarching goal of EUREC[4]A (Elucidate the Couplings Between Clouds, Convection and Circulation) is to improve our understanding of how trade-wind cumuli interact with the large-scale environment (Bony et al., 2017). Water vapour is one key ingredient of the atmospheric environment controlling the life cycle of convection with strong feedback on the large-scale

circulation and energy budget (Sherwood et al., 2010). The mechanisms involved are thought to play a significant role in the climate sensitivity and model diversity (Sherwood et al., 2014).

The EUREC[4]A field campaign was elaborated to provide relevant observations on the cloud properties and their atmospheric and oceanic environment across a range of scales (Stevens, 2021). The measurement platforms were deployed over two theatres of action. The first, referred to as the 'Tradewind Alley', extends from an open ocean mooring at 51° W, 15° N to the Barbados

Cloud Observatory (BCO), 59° W, 13° N (Stevens et al., 2016). The second named the 'Boulevard des Tourbillons' extends from 7° N, off the northern Brazil coast, roughly to the BCO. Most of the distant open sea water vapour measurements were made by Research Vessels (R/Vs) embarking radiosonde systems, lidars and microwave radiometers, for what concerns water vapour measurements. They were completed by aircraft platforms leaving from the Brigdetown, Barbados, airport, embarking dropsondes, in-situ and remote sensing measurement systems. The aircraft operated mainly in a 200-km diameter circle centred

on 57.7° W, 13.3° E in the Tradewind Alley, upstream from the BCO. In addition to the research instrumentation deployed on these platforms, ground-based and ship-borne GNSS receivers were operated to provide high temporal resolution Integrated Water Vapour (IWV) measurements. Results of ship-borne GNSS measurements from the R/Vs Meteor, Maria S. Merian, and Atalante are described in a companion paper (Bosser et al., 2020). This paper focuses on the ground-based GNSS network measurements.



The ground-based GNSS network comprises a total of 49 stations of which 47 are permanent instruments devoted primarily to geoscience investigations. Most of these stations belong to the Coconet continuoulsy operating GNSS network in the Caribbean (https://coconet.unavco.org/, last access: 29 January 2021) and provide raw observations as well as station positions and velocities for monitoring, understanding, and prediction of various geo-hazards (earthquakes, hurricanes, flooding, volcanoes, and landslides). Some other stations belong to the French permanent GNSS network, so-called 'Reseau GNSS

Permanent' (RGP) operated by IGN in the Caribbean and French Guyana territories. Figure 1 shows the map of the stations. Because of their location on the nearby Caribbean islands, most of the GNSS measurements mainly provide information on the large scale atmospheric environment downstream the Tradewind Alley and the Boulevard des Tourbillons. Closer to the main theatre of action of the field campaign, two collocated ground-based GNSS stations have been set up at the BCO in the framework of EUREC$^4$A. Two stations were installed to insure redundancy during the campaign. They complement the

permanent instrumentation operated at the BCO (Stevens et al., 2016). The BCO GNSS stations were installed on 31 October 2019 and are planned to operate for a full annual cycle at least. The measurements from these two stations are transmitted in near real time thanks to the BCO infrastructure and are processed operationally by IGN in the RGP processing stream, on one hand, and by ENSTA_B/IPGP (ENSTA Bretagne & IPGP), on the other hand. Both processing centers run continuously and provide IWV data with slightly different latency and precision. A more accurate research-mode processing was also set up by

ENSTA_B/IPGP to reprocess the data from all 49 stations in a homogeneous way, but for a limited period, from 1 January to 29 February 2020, hence including the Coconet and RGP stations. A first release of this data set is now available on the EUREC$^4$A data base hosted by the AERIS data center (Bock, 2020b).

       This paper aims at describing the ground-based GNSS data processing details and assessing the quality of the IWV retrievals from both the operational and research processing streams. Section 2 describes the GNSS data processing and IWV conver-

sion details. Section 3 compares the zenith tropospheric delay (ZTD) estimates and the IWV retrievals from the various data processing streams for the two BCO stations. A preliminary intercomparison with IWV data from other instruments operated at BCO, namely radiosondes and a microwave radiometer, is presented as well. Section 4 provides a validation of the ECMWF fifth reanalysis (ERA5) (Hersbach et al., 2020) at regional scale with respect to the IWV retrievals from the extended GNSS network and a preliminary analysis of the large scale IWV variations related to mesoscale cloud organisation in the Tradewinds.

Section 5 concludes the study.

## 2   GNSS data processing and IWV retrieval

A vast literature is covering the basic concepts of GNSS data processing for meteorological applications (see Guerova et al. (2016)). It will not be repeated here.

### 2.1   GNSS data processing streams

The GNSS measurements from the two BCO stations, hereafter called BCON and BCOS, are processed in four different operational streams. Two of the streams are run by IGN as part of the RGP operations as will be referred to as 'RGP NRT'





(Near Real Time) and the 'RGP daily'. Both consist in network processing solutions were the BCO stations are processed in baselines formed with about 40 permanent stations operated in the Caribbean and French Guyana region and beyond. The processing results, hereafter called "solutions", are produced on hourly and daily basis, respectively. The NRT solution uses

ultra-rapid satellite clocks and orbits, and processes the measurements in 6-hourly windows shifted by one hour every hour. For the permanent GNSS stations in the region, the ZTD estimates for the most recent hour are available within 45 min after the end of measurements and are distributed to Numerical Weather Prediction (NWP) centres via the EUMETNET-GNSS water vapour program (E-GVAP). Although the BCO stations were included in this stream they are not registered as permanent stations at E-GVAP and are thus not assimilated by the NWP centres. Due to the sliding window processing approach, each

hourly time slot is thus processed six times. The accuracy of the corresponding ZTD estimates depends on the time slot as will be illustrated in the next Section. The main reason is that the quality of the NRT clocks and orbits is lower in the more recent time slot because they are predicted (observations are not available in due time to constrain that part of the orbit calculation). The sampling of the ZTD estimates of the NRT solution is 15 min. On the other hand, the 'daily' processing stream is updated every day before 05 UTC. It is based on a longer time windows of 24 hours and benefits from slightly improved clock and orbit

products. Its accuracy is thus slightly higher. The sampling of the ZTD estimates of the daily solution is 1 h. This processing stream is not sent to E-GVAP but is rather devoted to an operational quality control of the observations from the RGP stations. It is also used for a posteriori verification of the NRT solution.

The data from the BCO stations were also included in two other operational streams operated by ENSTA_B/IPGP in the framework of ACTRIS-France (Aerosol, Cloud and Trace Gases Research Infrastructure – France, https://www.actris.fr/, last

access: 29 January 2021). These two streams are referred to as 'GIPSY rapid' and 'GIPSY final'. They differ only by the clock and orbit products which are released with daily and fortnightly latency, respectively. The GIPSY rapid processing is equivalent to the RGP daily processing, while the GIPSY final processing on the other hand provides a higher level of accuracy. Both streams provide ZTD estimates with a sampling of 5 minutes.

Apart from the usage of different clock and orbit products, the processing streams run by both analysis centres also differ

by their software packages and related processing options as summarised in Table 1. Most importantly, IGN uses the Bernese GNSS version 5.2 software in double-difference processing mode (Dach et al., 2015) while ENSTA_B/IPGP use the GIPSY OASIS II version 6.2 software in precise point positioning (PPP) mode (Zumberge et al., 1997). Various studies have compared the results and discussed the benefits and drawbacks of both approaches. When consistent clock and orbit products are used, there is generally a good level of agreement (i.e. a few mm RMS differences in ZTD estimates).

Finally, the regional network including 49 GNSS stations was reprocessed by ENSTA_B/IPGP using again GIPSY OASIS II version 6.2 software, in a very similar scheme as the final stream but with a few improved processing options. This processing is expected to be the most accurate. It is referred to as 'GIPSY repro1' in the following.





## 2.2 GNSS data quality checking

The quality of the data processing results can be checked with two types of information: 1) global information quantifying the
accuracy of the processing in the 6-hourly or 24-hourly time window (e.g. reduced sum of squares of residuals, percentage of
solved ambiguities) and 2) the formal errors of the estimated parameters (station coordinates and ZTDs).

For the operational streams, only the second type of information was available and a basic screening procedure was thus
applied in the form of a range check on ZTD estimates with limits [1 m, 3 m] and on for ZTD formal errors with a limit of 0.02
m.

For the GIPSY repro1 solution, a more elaborated quality check was made using both types of information for all stations.
Mean statistics over the 2-month period are reported in Table S1 of the supplemental material. The mean RMS of residuals
for all 49 stations ranged between 0.009 m and 0.015 m. Five stations with values above 0.013 m may be unreliable (N240,
DEHA, AIRS, BOUL, HABL). These stations have also small mean percentage of fixed ambiguities ($\leq 50\,\%$) and/or significant
temporal variations in both parameters. Station position repeatability was in general good both in the horizontal ($\leq 0.005m$)
and vertical ($\leq 0.015m$) except for two stations (BOUL, OLVN). In a second step, a tighter range check completed with an
outlier check were applied following the recommendations of Bock (2020a). The range check limits were [0.5 m, 3.0 m] on
ZTD values and 0.01 m on formal errors. The outlier check limits were computed for each station based on its median $\mu$ and
standard deviation $\sigma$ statistics over the 2-month period according to $[\mu - 5\sigma, \mu + 5\sigma]$ for the ZTD values and $[0, \mu + 3.5\sigma]$ for
the formal errors. This screening procedure rejected $0.21\,\%$ of the ZTD data due exclusively to the formal error outlier check.
The GIPSY ZTD time series are generally quite small with few outliers as confirmed by the low outlier detection rate.

## 2.3 GNSS ZTD to IWV conversion

The ZTD is the sum of the zenith hydrostatic delay (ZHD) and the zenith wet delay (ZWD):

$$ZTD = ZHD + ZWD \tag{1}$$

During the data processing, ZHD is fixed to an a priori value, either from an empirical model or from a NWP model analysis
(Boehm et al., 2007), and ZWD is estimated. Errors in the a priori ZHD can propagate to the estimated ZWD but, in general,
this error is small (< 1 mm) and the sum of both is very close to the real ZTD. When IWV is to be extracted, it is desirable to
use a more accurate ZHD value computed from the surface air pressure:

$$ZHD = 10^{-6} k_1 R_d \frac{P_s}{g_m} \tag{2}$$

where $k_1$ is the dry air refractivity coefficient, $R_d$ the dry air specific gas constant, $P_s$ surface air pressure, and $g_m$ the mean
acceleration due to gravity (Davis et al., 1985). IWV is derived from ZWD by applying a delay to mass conversion factor
$\kappa(T_m)$:

$$IWV = \kappa(T_m) \times ZWD \tag{3}$$





$\kappa$ is a semi-empirical function of the weighted mean temperature $T_m$ defined as (Bevis et al., 1992):

$$\kappa(T_m) = \frac{10^6}{R_v(k'_2 + \frac{k_3}{T_m})} \tag{4}$$

where $k'_2$ and $k_3$ are refractivity coefficients for the water molecule, $R_v$ is the specific gas constant for water vapour. $T_m$ is defined as:

$$T_m = \frac{\int \rho_v(z)dz}{\int \frac{\rho_v(z)}{T(z)}dz} \tag{5}$$

where $\rho_v(z)$ and $T(z)$ and the specific mass of water vapour and the air temperature, respectively, at height $z$ above the surface. The integrals are from the surface to the top of the atmosphere.

Practically, the GNSS IWV estimates are converted from the ZTD data using equations (1) to (4) with the auxiliary data, $P_s$ and $T_m$, given at the position of the GNSS station. The uncertainty in the IWV estimates due to errors in the auxiliary data can be assessed by the partial derivatives: $\frac{\partial IWV}{\partial ZTD} = -\frac{\partial IWV}{\partial ZHD} = 0.16 \,(\mathrm{kg\,m^{-2}}) \,\mathrm{mm^{-1}}$, and $\frac{\partial IWV}{\partial \kappa} = 0.20 \,(\mathrm{kg\,m^{-2}}) \,(\mathrm{kg\,m^{-3}})^{-1}$, or equivalently $\frac{\partial IWV}{\partial P_s} = 0.35 \,(\mathrm{kg\,m^{-2}}) \,\mathrm{hPa^{-1}}$ and $\frac{\partial IWV}{\partial T_m} = 0.12 \,(\mathrm{kg\,m^{-2}}) \,\mathrm{K^{-1}}$. The numerical values were computed for the average conditions of the 2-month period (January-February 2020), resulting in $\overline{\kappa} = 164 \,\mathrm{kg\,m^{-3}}$ and $\overline{ZWD} = 0.20$ m.

Various sets of refractivity coefficients and auxiliary data were available and used with our operational and research GNSS products and are described in the subsections below.

### 2.3.1 RGP NRT and daily results

This dataset is provided only for the BCON and BCOS stations which benefit from colocated surface air pressure and temperature measurements. The measurements come from Vaisala PTU200 Sensors connected directly to the GNSS receivers and are 160 included in the raw GNSS data files. The ZHD estimates are therefore computed using Eq. 2 and the $T_m$ values are computed using the widely used Bevis et al. (1992) formula:

$$T_m = 70.2 + 0.72T_s \tag{6}$$

where $T_s$ is the surface pressure. This formula was derived by Bevis et al. (1992) from a radiosonde data set over the USA with a RMS error of $4.7K$. The refractivity coefficients used for the computation of $ZHD$ and $\kappa$ are from Thayer (1974), the 165 thermodynamics constants are from ICAO (1993), and the $g_m$ formula is from Saastamoinen (1972).

### 2.3.2 GIPSY rapid and final results

This dataset is also provided only for the BCON and BCOS stations but it used the a priori values for $ZHD$ available during the data processing. The values were obtained from the Technical University of Vienna (TUV) as part of the VMF1 tropospheric parameters computed from the ECMWF operational analysis (Boehm et al., 2006). The TUV also computes $T_m$ values using 170 Eq. 5 from the same analysis. Both products are made available on a global grid with 2° latitude and 2.5° longitude resolution every 6 hours. The refractivity coefficients and $g_m$ are the same as for the RGP data set.



### 2.3.3 GIPSY repro1 results

For the reprocessed data set we used a more rigorous approach following Bock (2020a). Here the $ZHD$ and $T_m$ values are computed from ERA5 pressure-level data at the four surrounding grid points and are interpolated bilinearly to the positions of the GNSS antennas. The ERA5 fields were used with the highest temporal and spatial resolutions: 1-hourly and $0.25° \times 0.25°$, respectively. The refractivity coefficients and thermodynamics constants were updated from Bock (2020a) to account for the global $CO_2$ content for January 2020 (see the Appendix). The $g_m$ formula proposed by Bosser et al. (2007) was used instead of the Saastamoinen (1972) one.

This methodology provides the most accurate IWV estimates regarding the available data at the 49 stations of the extended network and the best knowledge of uncertainties due to the various empirical formulas ($T_m$ and $g_m$) and auxiliary data.

### 2.3.4 Uncertainty due to IWV conversion methods

The three data sets (RGP, GIPSY operational, and GIPSY repro1) used different conversion methods, which are associated with different uncertainties. Although repro1 is the most accurate, it is not an operational stream and covers only a limited period of time. Users may thus be interested by the near real time and extended time coverage of the two operational data sets available for stations BCON and BCOS. The consistency of the different data sets is assessed in the next Section for ZTD and IWV. Here we describe how the different error sources contribute to overall uncertainty in the IWV data.

Let us consider two ZTD solutions from two different processing streams, $ZTD_1$ and $ZTD_2$, converted to IWV using different $ZHD$ and $\kappa$ data, $ZHD_{1,2}$ and $\kappa_{1,2}$. The two IWV solutions, $IWV_1$ and $IWV_2$ write:

$$IWV_1 = \kappa_1 \cdot (ZTD_1 - ZHD_1) \tag{7}$$

$$IWV_2 = \kappa_2 \cdot (ZTD_2 - ZHD_2) \tag{8}$$

The difference $\Delta IWV = IWV_2 - IWV_1$ can be separated into systematic and random components and written as a function of the contributions (assumed independent) of the systematic and random differences of the three parameters $\Delta ZTD = ZTD_2 - ZTD_1$, $\Delta ZHD = ZHD_2 - ZHD_1$, $\Delta \kappa = \kappa_2 - \kappa_1$ as:

$$\overline{\Delta IWV} = \kappa_2 \cdot \overline{\Delta ZTD} - \kappa_2 \cdot \overline{\Delta ZHD} + \frac{IWV_1}{\kappa_1} \overline{\Delta \kappa} \tag{9}$$

$$(\sigma_{\Delta IWV})^2 = (\kappa_2 \cdot \sigma_{\Delta ZTD})^2 + (\kappa_2 \cdot \sigma_{\Delta ZHD})^2 + \left( \frac{IWV_1}{K_1} \sigma_{\Delta \kappa} \right)^2 \tag{10}$$

The magnitude of the actual errors in the auxiliary data used in the different conversion methods can be appreciated from Fig. 2. The Figure shows that the $ZHD$ data computed from ERA5 reanalysis and PTU sensor agree well. The mean pressure (ZHD) difference, ERA5 - PTU, amounts to $-0.13 \pm 0.20$ hPa ($-0.30 \pm 0.45$ mm), which impact on IWV represents $0.045 \pm 0.070 \, \mathrm{kg\,m^{-2}}$. ERA5 is thus a good source of surface pressure for the IWV conversion with negligible bias. The ZHD estimates computed from ECMWF by TUV on the other hand have significant aliasing errors (Figure 2). This is due to the strong atmospheric tide seen in the surface pressure that cannot be resolved by the 6-hourly analysis data (shown by crosses upon the red dashed line in the lower ZHD panel).





Regarding the $\kappa$ data, TUV and ERA5 agree well but the PTU-derived curve has significant errors because it uses the empirical formula given by Eq. 6 which is based on surface temperature and does not well correlate with the upper air variations

that influence $T_m$. In the case of the RGP data set, the use of Bevis et al. (1992) formula is the dominant error source in the IWV conversion process, while in the case of GIPSY rapid and final results, the use of 6-hourly ZHD data from TUV is the main error source.

We also evaluated the accuracy of the ERA5 $T_m$ estimates in comparison to $T_m$ computed from the radiosonde data at the BCO. The mean difference and standard deviation of the resulting IWV estimates for 138 soundings were $0.03 \ \mathrm{kg\,m^{-2}}$ (ERA5

- soundings) and $0.05 \ \mathrm{kg\,m^{-2}}$, respectively.

Compared to the differences in IWV estimates resulting from the use of different auxiliary data, the impact of the refractivity constants (Thayer (1974) vs. Bock (2020a)) is rather a small and represents a bias of $0.045 \ \mathrm{kg\,m^{-2}}$.

## 3   ZTD and IWV intercomparisons at the BCO

In this Section we inter-compare both ZTD and IWV results from the five processing streams. The motivation for comparing

the ZTD results is that it reflects the uncertainty due to the GNSS processing only while the IWV comparison includes the conversion errors discussed in the previous Section. The uncertainty in the ZTD data is of interest to the data assimilation community since ZTD are currently assimilated in most NWP models (rather than IWV). On the other hand, the uncertainty in the IWV data may be of interest to users who wish to analyse the IWV directly (e.g. for process studies, intercomparison with other observational techniques, or verification of atmospheric model simulations).

### 3.1   RGP NRT results

As explained above, in the RGP NRT solution, each hourly time slot is processed six times while it is shifted from the most recent position (referred to as NRT.6) to the oldest (NRT.1) in the 6-hourly time window. The highest precision is expected in the central time slots (NRT.2 to NRT.5) because these are farther from the edges and are best constrained by the observations.

#### 3.1.1   ZTD intercomparison

The overall uncertainty in the NRT solution is mainly due to the ultra-rapid satellite products and small window-size (impacting the ambiguity resolution). It can be evaluated by comparison with the RGP daily or any of the GIPSY solutions. The upper part of Table 2 shows the results for the ZTD estimates. The comparison to RGP daily reveals a trend in the mean difference from NRT.1 to NRT.6, with a large negative bias in NRT.1 of -0.04 m (RGP NRT ZTD < RGP daily ZTD). This bias was actually unexpected. It is an artifact due to the propagation of tropospheric gradient biases from one time slot to the next. This feature

was corrected in July 2020 and the bias is no longer present in the current operational NRT product. The standard deviation of differences shows a minimum (9.4 mm) for the NRT.4 solution. The values are actually smaller in the central time slots as expected. However, the formal errors (top row of Table 2) predict smaller uncertainty in the first time slot instead. This is a consequence of using the previous normal equations.



The comparison with GIPSY repro1 confirms these conclusions although there are small differences in the results: mainly,
the bias is slightly offset by 4 mm. Since this comparison involves two different software and processing approaches, these
small differences are not surprising. Note that the number of comparisons are also different by a factor of four due to the
different ZTD sampling of the three solutions. Compared to GIPSY repro1, the RGP NRT.4 has the smallest bias and standard
deviation. We recommend thus to use this solution for near-real-time applications when timeliness is not too restrictive (e.g. in
NWP assimilation schemes, the ZTD data from NRT.6 are used which are a bit less accurate).

Figure 3 shows the ZTD and formal error time series from the RGP NRT.4 solution and the other two processing streams. It
it seen that the temporal variations in ZTD are very consistent among all three data streams with small differences ($< 0.01$ m)
compared to the ZTD variations ( 0.1 m). On the other hand, the formal errors of the NRT solution are much larger than those
of the RGP daily and the GIPSY repro1 solutions. GIPSY repro1 has also more stable formal errors than RGP daily.

### 3.1.2 IWV intercomparison

The lower part of Table 2 shows the IWV results. In the case of the comparison of RGP NRT to RGP daily, the IWV results
are consistent with the ZTD results since both solutions use the same ZHD and K conversion data. The IWV differences are
here proportional to the ZTD differences with a factor of 162 $\mathrm{kg\,m^{-3}}$which is close to the mean value of K=164 $\mathrm{kg\,m^{-3}}$. On
the other hand, the comparison with GIPSY repro1 includes the differences in the conversion data, and the ratio of IWV to
ZTD results is not a constant factor. This point is further discussed in the next sub-section. Similar to the ZTD results, the IWV
solution for NRT.4 is in good agreement with RGP daily and GIPSY repro1.

Figure 4 compares the NRT.6 IWV estimates to GIPSY repro1 for the time period from 10 to 20 January 2020 marked by a
period of two days with high IWV around 45 $\mathrm{kg\,m^{-2}}$. It is striking that the NRT.6 solution shows spurious oscillations with a
period about 1 hour. This feature is due to the strong relative constraint (1 mm) in the NRT solution with an hourly update. The
constraint is the same in the NRT.4 slot but the oscillation is strongly damped (not shown). This result also militates for using
the NRT.4 solution rather than the NRT.6 solution when relevant.

### 3.2 RGP daily and GIPSY results

Similar ZTD and IWV comparisons have been performed for the other processing streams which are of interest for offline
applications.

The left part of Table 3 shows the ZTD comparison results. The comparison of RGP daily to GIPSY rapid and repro1 show
very similar results. The agreement between the two processing software is quite good, with a small difference in the mean
values of 2.1-2.4 mm for station BCON and 1.1-1.3 mm for station BCOS. The standard deviations of ZTD differences is about
5.7-5.8 mm for both stations. The differences reflect mainly the impact of using different satellite products, mapping functions,
tropospheric model constraints, and elevation cutoff angles. The comparison of the three GIPSY solutions show comparatively
much better agreement with almost no bias (the mean differences are around -0.1 to 0.3 mm) and very small standard deviations
(0.8 to 2.8 mm). The comparison of the two operational solutions (GIPSY rapid and final) shows the smallest standard deviation
because they use exactly the same processing options; the only difference is in the satellite products but they were produced by





the same analysis centre (JPL) and are highly consistent. The comparison of the GIPSY operational solutions to repro1 shows slightly larger differences which are due to small differences in the processing options.

IWV comparisons have been performed for the same five combinations. In addition to the ZTD differences due to the
processing strategies these comparisons will include the effect of the IWV conversion methods:

- RGP daily and GIPSY rapid differ by ZHD (PTU vs. TUV) and K (Bevis vs. TUV); strong impact is expected from differences in both parameters;

- RGP daily and GIPSY repro1 differ by ZHD (PTU vs. ERA5) and K (Bevis vs. ERA5); strong impact is expected from K;

- GIPSY rapid and final use the same conversion parameters;

- GIPSY rapid and final differ from repro1 by ZHD and K (TUV vs. ERA5); strong impact is expected from ZHD;

The right part of Table 3 shows the IWV results and confirms they include additional errors compared to the ZTD results (left part the Table). Most notably: (i) a change in the bias between RGP daily - GIPSY rapid and RGP daily - GIPSY repro1 because the two GIPSY solutions use different parameters. In the first comparison the bias is due to the ZHD data used in
the GIPSY rapid solution (TUV) and the K data in the RGP daily solution (due to Bevis et al. (1994) formula). In the second comparison, only the K bias in the RGP rapid solution is involved. (ii) a noticeable bias and standard deviation between the two operational GIPSY solutions and the repro1 due to the ZHD data used in the former (TUV). The two operational GIPSY solutions are highly consistent in IWV (similar to the ZTD solutions) but this result hides the fact that they include common ZHD errors.

Table 4 quantifies more precisely the contributions from the difference error sources (ZTD, ZHD, and K) to the total IWV differences, where the mean and random components have been separated according to Eqs. 9 and 10. It can be seen that:

- In the RGP daily and GIPSY rapid comparison, all three error sources contribute in the same proportions to the bias, while the random errors are dominated by ZTD errors.

- In the RGP daily and GIPSY repro1 comparison, the small bias in RGP daily is a result of almost exact cancelling of a
ZTD bias and a K bias in the RGP data. The random errors in the RGP data are again dominated by ZTD errors.

- In the GIPSY final and repro1 comparison, the bias in the GIPSY final is almost only due to a bias in ZHD, while the random errors are due to ZTD and ZHD differences of the same magnitude.

In conclusion, compared to the GIPSY repro1 dataset which uses the most accurate data processing and conversion parameters, both operational streams show small systematic and random errors on the level of $\pm$ 0.5 $\mathrm{kg\,m^{-2}}$ and $\pm$ 1.0 $\mathrm{kg\,m^{-2}}$,
respectively.



## 3.3 GNSS compared to other IWV data sources

Two other instruments operated at the BCO facility are used here: the Vaisala RS41/MW41 radiosonde system (Stephan et al., 2020) and the RPG, HATPRO-G5, microwave radiometer (MWR) (Rose et al., 2005). The IWV retrievals from these systems were compared to IWV estimates from the BCON GNSS station (GIPSY repro1) and ERA5 IWV data for the period from 1 January to 29 February 2020. For the radiosondes, the level-1 pressure (P), temperature (T), and relative humidity (RH) data were used and IWV was computed as:

$$IWV_{RS} = \int_{P_{GPS}}^{P_{top}} q(P)/g(P)dP \tag{11}$$

where the integral extends from the GNSS station height to the top of the sounding ( 24 km on average), $q(P)$ is the specific humidity computed from RH and T using Tetens (1930) saturation pressure formula over water, $g(P)$ is the acceleration of gravity as a function of altitude. The 1-sec time sampling of the radiosonde data provide high vertical resolution (about 5 m in the lower troposphere). They were checked for consistency and thinned for including only increasing altitude points. Very few data gaps were noticed in the vertical profiles which confirms that the sounding operations worked fine all along the campaign and did not need further correction or screening. The BCO soundings provided a nearly continuous sampling of the upper air conditions between 16 January and 19 February, every 4 hours (Stephan et al., 2020). The MWR worked for a more extended period from 1 January to 15 February. The brightness temperature measurements were used to retrieve IWV using a neural network algorithm provided by the manufacturer. In precipitating conditions, the measurements usually experience strong biases due to wet radome emissions (see Fig. 5) and are screened out according the a rain detection index (including also sea spray detection). The IWV contents are nominally retrieved with a 1-sec sampling which we down-scaled to 5-min, by computing arithmetic means to be consistent with the GNSS sampling. Some outliers remained in the MWR IWV series which were subsequently removed from the comparisons by a simple outlier check of the IWV differences with limits mean $\pm$ 3 standard deviations).

The ERA5 IWV contents above the GNSS station were computed from the hourly pressure level data at the four surrounding grid points and interpolated bilinearly to the position of the GNSS antenna. The use of pressure level data instead of model level data induces a minor bias of $0.2 \pm 0.1$ $\mathrm{kg\,m^{-2}}$ but these data are more convenient to use and consistent with the Tm computation (see Sect. 2.3.3).

In addition, we also included the twice-daily (00 and 12 UTC) radiosonde measurements from the operational radiosonde station at Grantley Adams International Airport (GAIA, WMO code 78954) located 11 km away from the BCO. This station used GRAW DFM-09 at the time of the campaign (Kathy-Ann Caesar, personal communication). The sonde data were retrieved from the University of Wyoming sounding archive and contained on average $85 \pm 6$ vertical levels up to an altitude of $29 \pm 2$ km. The vertical sampling of these data is coarser than the BCO data but fine enough to compute proper IWV contents. The latter were computed in a similar way as for the BCO soundings except near the surface where the ERA5 pressure level data





were used to complete the sounding profiles. This is because the GAIA station is located at an altitude of 57 m, slightly above the BCO which is at 25 m altitude.

Figure 5 shows the IWV time series of the different data sources at their nominal time resolutions. The period underwent large moisture variations with IWV fluctuations between 20 and 55 $\mathrm{kg\,m^{-2}}$. Several remarkable situations are observed such as the peaks on 22 January around 12 UTC and 24 January around 00 UTC, and a few more in February (days 40, 42, 43-44, 45). The agreement between the five data sets is good in general but ERA5 is seen to underestimate the IWV during the aforementioned peaks. Quite obvious is also the bias between the two sounding data, with the BCO IWV contents systematically higher than the GAIA measurements. Less easy to distinguish, but nevertheless significant, is the offset in the HATPRO MWR IWV retrievals before and after the 27 January. The HATPRO data are actually more consistent with the GAIA data before that date and instead more consistent with the BCO radiosonde data after. Analysis of the measured brightness temperatures showed that during the former period, the measurements were less accurate due to an instrumental malfunctioning which was fixed on 27 January.

Figure 6 compares the IWV data from the five sources pairwise and includes some consistency statistics. The slope and offset parameters were fitted using the York et al. (2004) method to account for errors in both coordinates. For both radiosondes and HATPRO the uncertainty in the IWV estimates was assumed to be 5 %. For ERA5 the uncertainty was computed as the max - min of the IWV values at the four surrounding grid points before the horizontal interpolation, which is a measure of the representativeness error (Bock and Parracho, 2019). The uncertainty estimates over the study period were $1.71 \pm 0.31 \mathrm{~kg\,m^{-2}}$ for the BCO soundings, $1.57 \pm 0.28 \mathrm{~kg\,m^{-2}}$ for the GAIA soundings, $1.65 \pm 0.25 \mathrm{~kg\,m^{-2}}$ for HATPRO, and $1.48 \pm 0.86 \mathrm{~kg\,m^{-2}}$ for ERA5. For GNSS, the ZTD formal errors were converted to IWV and rescaled with a factor of 5 in order to be consistent with the other data sources, yielding a final GNSS IWV uncertainty of $1.15 \pm 0.16 \mathrm{~kg\,m^{-2}}$.

Results from the GPS, the BCO and GAIA sondes, and the ERA5 comparisons are reported in Fig. 6. While IWV varies from 20 to 55 $\mathrm{kg\,m^{-2}}$ over the period, the bias between the two sondes is -2.89 $\mathrm{kg\,m^{-2}}$ (GAIA - BCO) with a slope of 0.94 and an offset of -0.69 $\mathrm{kg\,m^{-2}}$. The GAIA data are drier than the BCO data over the full observation range. The comparison of profiles shows that the humidity measurements from the two sondes differ mainly in the lowermost 2.5 km where the mean difference in $q$ is larger than 1 $\mathrm{g\,kg^{-1}}$. One contribution to this difference may be the difference of trajectories of the sounding balloons released from the two sites. Balloons released from the BCO site usually travel west and southwards over the Barbados island until they reach an altitude of 6-8 km when they enter the westerly jet. The balloons released from GAIA drift in similar directions but arrive earlier over the open sea as they are released from the Southern part of the island. Since the humidity over the sea is larger, the GAIA measurements would actually be expected to be moister. However, this reasoning is neglecting moisture transport associated with land and sea breezes. The moisture observations at the BCO exhibit actually a day/night variation in IWV of 1.7 $\mathrm{kg\,m^{-2}}$ whereas the variation at GAIA is significantly smaller (1.1 $\mathrm{kg\,m^{-2}}$). A similar "island effect" was previously evidenced in Colombo, Sri Lanka, where the land/see breezes contributed to a daytime boundary layer moistening and nighttime drying observed in radiosoundings (Ciesielski et al., 2014a). Although this differential diurnal variation is not negligible, it is not large enough to explain the mean bias between the BCO and GAIA sondes. Instead we suspect the difference in sonde types to play a more central role.





The Vaisala RS41 sondes used at BCO are from the last generation of sondes and are considered to provide significantly improved temperature and humidity measurements compared to previous sonde types (e.g. Vaisala RS92) especially in cloudy conditions (Jensen et al., 2016). High confidence in the BCO soundings suggests that the GAIA soundings have a significant

dry bias which may impact humidity analyses from NWP models that assimilate these observations. This seems to be the case for ERA5 and would explain the high consistency between ERA5 and GAIA IWV estimates (bias of 0.29 $\mathrm{kg\,m^{-2}}$ and near unity slope) and the large difference of ERA5 compared to the BCO (bias of -2.33 $\mathrm{kg\,m^{-2}}$ and slope of 0.93). The GNSS IWV estimates are intermediate between the two sounding results. Using GNSS as the reference, we conclude on a wet bias in the BCO radiosonde data of 1.64 $\mathrm{kg\,m^{-2}}$ and a dry bias in the GAIA radiosonde data (-1.23 $\mathrm{kg\,m^{-2}}$) as well as in the ERA5 data

(-0.77 $\mathrm{kg\,m^{-2}}$). On the other hand, using the BCO soundings data as a reference one would conclude on a dry bias in all other IWV estimates, except the HATPRO MWR after the instrumental fix on 27 January, where the MWR retrieval is slightly moist. Since no reference water vapour measurements were made at the same time, it is difficult to establish the absolute sign of these biases. The GNSS - RS41 bias of -1.64 $\mathrm{kg\,m^{-2}}$ observed here represents a fractional bias of 5 % which is slightly larger the uncertainty of both systems and thus needs to be explained. From our previous experience comparing Vaisala RS92 sondes and

GNSS IWV estimates (produced using the same approach as in this study) we observed slightly smaller biases of ± 0.5 to 2 % (Bock et al., 2016). Further investigation is needed in the case of the RS41 vs. GNSS comparisons.

The comparison of HATPRO IWV retrievals has been separated in two batches (before and after 27 January). Compared to GNSS, the bias before (after) is -1.28 (+2.06) $\mathrm{kg\,m^{-2}}$ with a slope significantly larger than one indicating that the bias increases with the amount of IWV. Similar slope values are obtained in the comparison to the other data sources, which might

be attributed to the MWR training data set (Rose et al., 2005). Occurring biases could be further corrected by applying a clear-sky brightness temperature offset correction based on sounding data. In terms of temporal variability, the MWR, GNSS, and BCO sondes are in good agreement (standard deviation of differences ∼ 1 $\mathrm{kg\,m^{-2}}$ and correlation coefficient ≥ 0.98). The bias with respect to the BCO soundings is 0.48 $\mathrm{kg\,m^{-2}}$ (MWR too moist) during the second period, but this bias is within the known uncertainties.

ERA5 is biased low compared to both GNSS and BCO sondes with slopes lower than one, meaning that the dry bias increases at larger IWV values. The scatter plots in Fig. 6 exhibit a few outlying values which correspond to the situation when ERA5 underestimates the IWV peaks during the period 22-24 January (see Fig. 5). Quite surprisingly, the GAIA soundings during this period, although slightly drier, are much closer to GPS and BCO than ERA5. Inspection of the vertical humidity profiles (Figure 7) shows that ERA5 is close to the GAIA observations in the lower 2-2.5 km but does not account for the vertical

extension and the sharp drop in humidity at the top of the deep moist layer around 3-4 km on 22 January and around 5 km on 24 January. The mis-representation of the relative humidity profile is spectacular on 24 January 00 UTC. This might be due to the assimilation of other data (e.g. satellite humidity sounders) that are biased low in the mid-troposphere during this event. Inspection of radar reflectivity measurements from the BCO reveals that on both dates heavy rain was occurring over an extended part of the lower troposphere (up to 3.5 km height on 22 January around 10-11 UTC and 5.0 km on 23 January

around 22-23 UTC). This would explain the saturated air (RH=100 %) below 3 km for the former and below 5 km for the latter of the two BCO soundings. The GAIA soundings mimic the BCO soundings, but with a dry bias within these rainy layers.





Above this layer, it seems that the GAIA sondes have a moist bias, probably due to rain contamination of the humidity sensor, a problem that is corrected in the design of the newer RS41 sondes (Jensen et al., 2016). Above the moist layers, ERA5 fits well the BCO profiles.

Complementary statistics are reported in Table 5 where the data have been time-matched. The conclusions are essentially the same as discussed above, but the pair-wise biases can be more easily compared and combined.

## 4 Spatio-temporal distribution of IWV at the regional scale

### 4.1 Comparison of IWV retrievals from GNSS network and ERA5 reanalysis

Figure 8 shows the mean and standard deviation of hourly IWV from ERA5 and GNSS for the two-month period. The mean
IWV is seen to be relatively uniform over the region, with a small SW-NE gradient correlated with the sea surface temperature (SST) gradient (SST decreasing towards the NE). The maximum, around 35 $\mathrm{kg\,m^{-2}}$, is observed below 13° N in the more tropical part of the domain, and the minimum to the NE is around 30 $\mathrm{kg\,m^{-2}}$. The mean ERA5 field is generally in good agreement with the GNSS observations over the Caribbean Arc (CA), with an exception over Puerto Rico (18° N, 67° W) where ERA5 has a small dry bias (0.5 - 1 $\mathrm{kg\,m^{-2}}$). On average over all stations, the mean difference (ERA5 - GNSS) is
-0.30 $\mathrm{kg\,m^{-2}}$, pointing to a small dry bias of ERA5 in the region. This result is consistent with the bias that is discussed in the previous Section based on BCO results, and also with the results of a previous study (Bosser and Bock, 2021). But further investigation is needed to explain the reason of this dry bias in ERA5 (e.g. its link with the data assimilation).

In terms of temporal variability, the agreement between ERA5 and GNSS is also quite good, except over Puerto Rico. There is significant spatial modulation of the magnitude of variability. Again the maximum is observed to the South over the tropical
Atlantic Ocean. The average difference of standard deviation (ERA5 - GNSS) is -0.25 $\mathrm{kg\,m^{-2}}$, meaning that the variability in ERA5 is slightly underestimated compared to GNSS. This may be partly due to a difference in representativeness between the GNSS point observations and the gridded reanalysis fields (Bock and Parracho, 2019) and partly to some special situations where the reanalysis underestimates high IWV contents (see Fig. 5).

Table S2 in the Supplement gives the comparison results for all stations. The mean and standard deviation of IWV differ-
ences (ERA5 - GNSS) have been cross-compared with the GNSS data quality diagnostics from Table S1 and no significant correlation was found. We therefore believe the main differences are not due to GNSS uncertainties but rather to differences in representativeness such as evidenced particularly over Puerto Rico. Closer inspection of the ERA5 orography for this island shows that the topography is largely mis-represented in the model where the highest elevation is 316 m above sea level whereas the real topography reaches 1338 m. Also, the latitudinal extension of the island is exaggerated (e.g. GNSS station EMPR at
18.47° N is at an altitude of 10 m whereas the nearest model grid point at 18.5° N is at 103 m). Since almost all the stations considered in this study are located on small volcanic islands with steep topography, the mis-representation of the topography is a major source of uncertainty in the GNSS and ERA5 comparison. This poses also some problems to the assimilation of observations taken from surface meteorological stations and upper air soundings.





## 4.2 Couplings between clouds, circulation, and humidity at the synoptic scale

To illustrate the spatio-temporal variability of IWV at the regional scale we have selected five GNSS stations representative of the different parts of the Caribbean Arc (CA). The GNSS station CRO1 (St Croix, US Virgin Islands, 17.76° N, 64.58° W) was chosen as representative of northern CA. The GNSS station of LDIS (Guadeloupe, 16.30° N, 61.07° W) and FFT2 (Martinique, 14.60° N, 61.06° W) were chosen as representative of central CA, GNSS station BCOS (13.16° N, 59.42° W) was selected as being representative of Barbados and the GNSS station of GRE1 (Grenade, 12.22° N, 61.64° W) was selected for southern

CA. The selection accounted for the continuity of the series (ideally 1440 hourly IWVs over the two months of January and February 2020), and the processing quality diagnostics (see Sect. 2.3.3 and Table S1). For the islands associated with several stations (e.g. 15 for Guadeloupe and 6 for Martinique), the GNSS station with the most complete IWV series and the best data quality was chosen.

The IWV times series associated with the five GNSS stations are shown in Fig. 9. They highlight substantial variability
in the course of the January-February 2020 period, alternating moist (in excess of 50 $\mathrm{kg\,m^{-2}}$) and dry (below 20 $\mathrm{kg\,m^{-2}}$) episodes, sometimes within a few days as for instance in Barbados where GNSS-derived IWV was observed to decrease from 54.3 $\mathrm{kg\,m^{-2}}$ to 17.8 $\mathrm{kg\,m^{-2}}$ between 2300 UTC on 23 January and 2100 UTC on 27 January 2020 (Figure 9d). Unsurprisingly, the time series of the five GNSS stations spanning over a region of 6° in latitude do not show obvious correlations, suggesting that they are not influenced by the same IWV-impacting weather at the same time in the course of the two months.

Among the processes likely to strongly impact the IWV fields in the Tradewinds, shallow convection is of paramount importance. Stevens et al. (2020) have shown that cloud mesoscale organisation in the Tradewinds is dominated by four main patterns referred to as Fish, Flowers, Gravel and Sugar. These patterns have also been shown to depend on environmental conditions (Bony et al., 2020; Rasp et al., 2020; Schulz et al., 2021). The clouds embedded in these patterns are characterized by different vertical and horizontal extensions, reflectivity, separation, etc. For instance, it has been shown that the clouds
compositing the Fish, Gravel, and Flowers have similar vertical extent (see Stevens et al. (2020), their Fig. 9, based on radar observations in Barbados), but different from the small clouds composing the Sugar. For details on the characteristics of the four cloud patterns, the reader is referred to Stevens et al. (2020); Schulz et al. (2021).

In order to obtain a first assessment of the IWV values characterizing the environment of the mesoscale cloud organization in the region of the CA, we have performed a visual classification of the cloud scenes over sixty days (between 1 January and
29 February 2020) over a domain spanning from 57.4° W to 67.6° W and from 8.9° N to 19.1° N. Our domain is similar in size to the one used by Stevens et al. (2020) for the same months between 2007 and 2016, but shifted west by 10° and slightly south as well to include the CA. The classification was performed by visual inspection of the MODIS Aqua and Terra visible images at 1330 and 1030 local equator crossing time available from NASA WorldView (https://worldview.earthdata.nasa.gov/, last access: 29 January 2021). Unlike Stevens et al. (2020), we did not classify the dominant cloud scenes across the domain, but
came up with a classification of the cloud scenes around each of the selected GNSS stations in order to more accurately link GNSS-derived IWVs with cloud organisation. This was necessary as on some days the different parts of the domain were not under the influence of the same cloud pattern as for example shown in the Aqua and Terra MODIS images on 19 January 2020





(Figure 10). On that day, St Croix was under the influence of a Fish while Barbados was surrounded by clear air. The clear-sky band in between two cloud clusters is actually part of the Fish pattern. Guadeloupe located off the southern edge of the Fish

structure and was surrounded by Gravel, while Martinique and Grenade further south were bordered by Sugar. On other days such as 13 January 2020, Gravel was observed to be uniformly spread across the CA domain (Figure 10). The classification performed for each of the five sites in January and February 2020 are provided in the Fig. 11.

For each of the five GNSS stations representative of the different parts of the CA, the time series of cloud classification is also shown, with Fish appearing in red, Gravel in green, Sugar in light blue, Flowers in dark blue and cloud-free conditions

in white. As also observed by Stevens et al. (2020) between 2007 and 2016, Gravel-type was the dominant mode of cloud organisation over the CA during January and February 2020, with a number of cases ranging from 19 in Guadeloupe to 27 in Grenade (Table 6). Fish was the next most dominant pattern of cloud organisation with a number of cases ranging from 10 in Martinique to 19 in St Croix and Guadeloupe. Sugar was the least observed cloud pattern (between 0 and 6 cases depending on the site) as also demonstrated by Stevens et al. (2020). Flowers were observed more often in the central and southern part

of the CA (between 8 to 11 days), than in the northern part (on 3 days only). Finally, Cloud-free cases in between Fish and Flowers were also observed on more than 10 days during the period of interest, except in the southern part of the CA where only a few cloud-free days are observed.

From the time series of IWV and the cloud classification shown in Fig. 9, and if we consider the two most dominant modes, the picture emerges that Fish pattern (in red) is more systematically associated with higher IWV values than the Gravel pattern

(in green). This visual impression is confirmed when computing the average IWV values associated with Fish and Gravel patterns over the five stations (Table 6). The mean IWV values in Fish environment are found between 40.5 and 35.1 $\mathrm{kg\,m^{-2}}$, while their counterparts in gravel environment range between 33.1 and 30.8 $\mathrm{kg\,m^{-2}}$. Differences in the mean Fish-related and Gravel-related IWV values range between 4.3 $\mathrm{kg\,m^{-2}}$ (Guadeloupe) and 7.4 $\mathrm{kg\,m^{-2}}$ (Grenade), and are found to be significant using a Student's t-test (see Table 7). Clear scenes over the islands of interest are also seen to be associated with rather dry

conditions, whereas Flowers are associated with intermediate moister conditions with mean IWV values ranging between those of Fish and Gravel. The differences in the mean Fish-related and clear conditions IWV values are also found to be significantly different (the difference ranging between 8.4 $\mathrm{kg\,m^{-2}}$ in St Croix and 5 $\mathrm{kg\,m^{-2}}$ in Guadeloupe). Student's t-tests also reveal that the difference in the IWV means of all the other pairs (i.e. Fish-Flowers, Gravel-Flowers, Gravel-Clear conditions and Flowers-Clear conditions) are not significant.

From our analysis it also appears that ambient conditions in Grenade are moister than in the rest of the CA (this is reflected in both the Fish-related and the Gravel-related mean IWV values), which is likely connected to the proximity of the Intertropical Converge Zone located over the northeastern part of South America in boreal winter. Interestingly, the driest conditions are observed in Guadeloupe (the northern part of central CA) which may be an indication of a more pronounced influence of the mid-latitudes.

In summary, based on a compilation of IWV values gathered from representative GNSS stations across the CA, we found that the environment of Fish cloud patterns to be moister than that of Flowers cloud patterns which, in turn, is moister than the environment of Gravel cloud patterns. This is consistent with the relative humidity profiles composited by Schulz et al. (2021).





Since Fish patterns are associated with weak winds (relative to Flowers or Gravel (Bony et al., 2020), it means that this high
humidity is related to mass convergence within the column, associated with ascent. The differences in the IWV means between
Fish and Gravel were assessed to be significant. Finally, the Gravel moisture environment was found to be similar to that of
clear, cloud-free conditions. The moister environment associated with the Sugar cloud pattern has not been assessed because
it was never prominent around the GNSS stations during the first two months of 2020 (but Sugar-like clouds occur very often
within mesoscale cloud patterns).

In the following, we focus on the specific period during which a large variation of IWV was observed at the GNSS station
BCOS in Barbados from $54.3 \ \mathrm{kg \, m^{-2}}$ to $17.8 \ \mathrm{kg \, m^{-2}}$ between 2300 UTC on 23 January and 2100 UTC on 27 January 2020,
associated with a transition from a Fish cloud pattern to clear, cloud-free conditions (see the cloud scene classification in Fig.
9d). For the period from 20 to 30 January, the extreme IWV values were indeed observed in Barbados (i.e. values given above),
the closest lowest values being observed in Martinique on 20 January and the closest highest value in St Croix on 24 January
2020.

On 22 January, a large southeast-northwest oriented Fish feature is observed with MODIS to extend over a large portion of
the CA, from Barbados to Guadeloupe, also covering Martinique (Figure 13a), while Grenade is located in clearer air south of
the Fish and St Croix is covered by another distinct Fish feature further north. Figure 14a shows the ERA5-derived IWV field
over the same domain as the MODIS Aqua visible image at roughly the same time (∼1500 UTC) together with the GNSS-
derived IWV values (overlain within open white circles at the location of the 49 GNSS stations). The Fish feature extending
over central CA is associated with a plume of rather high IWV over Barbados and Martinique in the ERA5 field, but does
not reach to Guadeloupe whereas GNSS retrievals indicate higher IWV values in the southern part of the island. The GNSS
stations in Barbados, Martinique and Guadeloupe all show IWV values in excess of $35 \ \mathrm{kg \, m^{-2}}$ after 1200 UTC on that day,
while lower values are observed in Grenade to the south and St Croix to the north (Figure 12). In Barbados, a maximum IWV
value of $48.6 \ \mathrm{kg \, m^{-2}}$ was observed at 1200 UTC which is associated with a deep moist lower troposphere as observed from
the radiosounding measurements made from the BCO (Figure 7a and b). Using trajectory analyses from the LAGRANTO
Lagrangian analysis tool (Sprenger and Wernli, 2015), Villiger et al. (2020) also evidenced that air parcels arriving at 1000-
700 hPa above the BCO are transported from high latitudes towards the BCO by an extratropical surface cyclone/upper level
trough located off the US East coast. The initially dry air parcels descend from upper-levels into the boundary layer, where
they experience a rapid moistening, before arriving at the BCO as anomalously humid.

On 23 January, a southeast-northwest oriented Fish feature is also observed with MODIS to extend between Barbados and
Martinique. Guadeloupe is under the influence of another distinct Fish feature further north, while Sugar is observed to surround
Grenade and cloud-free conditions are found over St Croix (Figure 13b). Three distinct IWV plumes are seen in the ERA5 field,
one over Puerto Rico, one further east over the Netherlands Antilles almost reaching Guadeloupe and one extending from the
southeast over Barbados and Martinique (Figure 14b). Drier air masses are seen in St Croix (located between two IWV plumes
to the north) and in Grenade along the southern edge of the southernmost plume. Comparison between ERA5 and GNSS
IWV values suggest that the southernmost plume in ERA5 is located a bit too far south as ERA5 IWVs are underestimated in
Barbados and Martinique, and slightly overestimated in Grenade. On that day also, the GNSS stations in Barbados, Martinique





and Guadeloupe all show IWV values in excess of $35 \text{ kg m}^{-2}$ after 1200 UTC, while lower values are observed in Grenade to the south and St Croix to the north (Figure 12). The maximum IWV value of $54.3 \text{ kg m}^{-2}$ observed at the end of the day in

Barbados is in good agreement with that derived from the high-resolution radiosounding performed at 2240 UTC at the BCO (Figure 5). It exhibits a deep moist layer, with a nearly $100 \%$ relative humidity in the first 5 km of the troposphere (Figure 7c and d) which was also highlighted by Stephan et al. (2020), their Fig. 9. Like on 22 January, the moist air below 5 km above the BCO is associated with transport from high latitudes towards the BCO by the extratropical surface cyclone/upper level trough off the US East Coast (Villiger et al., 2020).

On 24 January, the cloud scene over the CA is dominated by Fish features again, with the previously observed southern southeast-northwest oriented Fish pattern being shifted to the west, clearing most of the central CA and Barbados, while moving over Grenade (Figure 13c). It merges over the Caribbean Sea with a southwest-northeast oriented Fish structure going across the CA north of Guadeloupe. St Croix is underneath a distinct Fish at that time. As a result, Guadeloupe, Martinique and Barbados are in a mostly cloud-free air mass moving in westward behind the '<'-shaped Fish structure. As shown in Fig.

14c, the '<'-shaped structure of Fish pattern also reflects in the ERA5-related IWV field, with higher IWV values located to the south of Barbados and to the east of Grenade. The northern branch of the '<'-shaped plume appears to be located too far east in ERA5 as suggested by the slight overestimation of IWVs over Guadeloupe compared to the GNSS retrievals. Consistent with the above described spatial distribution, the GNSS stations in Grenade shows the highest IWV values (reaching $50 \text{ kg m}^{-2}$) after 1200 UTC, while lower similar values (and a similar decreasing trend) are observed in Barbados, Martinique, Guadeloupe

and St Croix further north (Figure 12).

On 25 January, Grenade and St Croix appear beneath the remains of two distinct Fish features, while cloud-free conditions are observed in the rest of the CA. The central part of the CA is bordered with Sugar to the east (Figure 13d). The '<'-shaped feature of higher IWVs is still visible in the ERA5 field further west with respect to the previous day, with most of the central part of the CA being located in a drier environment (Figure 14d). On that day also, the northern branch of the '<'-shaped IWV

plume appears to be located too far east in ERA5. The GNSS stations in Grenade shows the highest IWV values after 1200 UTC even though IWV is observed to decrease on that day as the '<'-shaped plume is moving west. IWV values in Barbados, Martinique and Guadeloupe are even smaller than the previous days as drier air masses continue to move in from the Tropical north Atlantic (Figure 12). A maximum of IWV is observed in St Croix in relationship with the presence of a Fish feature.

On 26 January, IWV values in Grenade, Barbados and Martinique continue to drop (Figure 12) as large scale cloud features

(and related higher IWV values) continue to be advected westward (Figure 13e and 14e, respectively). Cloud-free conditions now dominate the central part of the domain, while Fish features are observed in the northwestern most corner of the domain, covering both St Croix and Guadeloupe. As a result, the GNSS station in St Croix highlights a maximum of IWV reaching $50 \text{ kg m}^{-2}$, while larger IWV values than the previous day are also observed in Guadeloupe (Figure 12), consistent with the spatial distribution of IWV obtained with ERA5 (Figure 14e).

Finally, on 27 January, a well-defined Fish feature is observed in the north of the domain, covering the US Virgin Islands and St Croix (Figure 13f), which is associated with a plume of IWV values between 40 and $50 \text{ kg m}^{-2}$ (Figure 14f). All the other stations to the south are located in the drier, mostly clear-free air mass to the east of the Fish pattern. Martinique



and Guadeloupe are surrounded by Sugar while the environment of Barbados and Guadeloupe is observed to be cloud-free (Figure 11 and Fig. 13f). The GNSS station in St Croix highlights a IWV maximum of 50 $\mathrm{kg\,m^{-2}}$ for the second consecutive
day, while in Barbados and Grenade the GNSS stations record the lowest IWV values of the period after 1200 UTC as the driest conditions are evidenced with ERA5 over the Tropical North Atlantic east of central CA (Figure 14f). In Martinique and Guadeloupe, higher IWV values are observed (between 30 and 35 $\mathrm{kg\,m^{-2}}$) after 1200 UTC (Figure 12) as both stations appear to be located in regions of IWV gradients west of the drier air masses (Figure 14f).

It is also worth noting that from 16 to 22 January the IWV features as modelled with ERA5 are moving westward across the
CA domain (not shown). On 22 and 23 January, IWV features remain quasi-stationary, while from 24 January onward, there is a clear decoupling between the northern and southern part of the domain, with IWV features north of 14° N being advected eastward and IWV structures south of 14° N moving westward. This is the origin of the '<'-shaped IWV feature seen on 24 and 25 January (Figure 14c,d) as well as the the '<'-shaped Fish pattern observed in the MODIS images (Figure 13c,d). The eastward motion north of 14° N on 24 and 25 January is likely related to the growing influence of an extratropical disturbance
(sea surface pressure of 1005 hPa, not to be confused with the extratropical cyclone off the US East Coast) that formed north of Puerto Rico on 22 January, with its center located at 30° N, 65° W, and moved northeastward in the following days (the center of the disturbance is located at 35° N, 45° W on 25 January).

## 5 Conclusions

This paper describes the data processing streams and discussed the quality of GNSS ZTD and IWV retrievals from four
operational streams run by IGN and ENSTA-B/IPGP and one research stream (GIPSY repro1), see Table 1. The two operational streams run by IGN (RGP NRT and RGP daily) provide data for stations BCON and BCOS, the two GNSS stations installed at the BCO for EUREC[4]A in October 2019, as a well as a few other sites in the French overseas territories of the Antilles. The RGP NRT stream provides ZTD estimate within 45 min after the end of measurements which are generally available for assimilation into NWP models. We showed that this stream had some instabilities, especially in most recent 1-hour time slot,
which are corrected in the t-2h time slot. This instability was found to be related with the processing scheme and was corrected since then by IGN. The other operational streams were shown to be in good agreement, with a small bias of 2 mm in ZTD between the IGN and ENSTA-B/IPGP solutions due to the use of two different software packages. IWV estimates are made available for stations BCON and BCOS from all four operational streams since 31 October 2019. This offers the possibility to analyse the IWV time series both in near real time and on the long term. However, due to the use of various different ZTD to
IWV conversion methods and auxiliary data, the uncertainty in the IWV estimates is variable and not optimal in these streams (see Table 4). It is planned in the near future to improve and homogenize the operational IWV conversion approach for all four operational streams and make these data available to the users.

The GIPSY repro1 research stream includes 49 GNSS stations covering the Caribbean Arc (see Fig. 1). The ZTD and IWV estimates from this stream, which is the main focus of this paper, have been analysed for the period 1 January-29
February 2020 and made available with 5 minute time sampling for scientific applications. This stream used a slightly improved





processing approach and IWV conversion method and auxiliary data with reduced uncertainties. The uncertainty due to the IWV conversion is believed to be at the level of $\pm$ 0.2 $\mathrm{kg\,m^{-2}}$ (see the Appendix). The data have been thoroughly quality checked and screened for outliers (see the Supplemental material).

The GNSS IWV estimates set have been compared to the Vaisala RS41 radiosonde measurements operated from the BCO and to the measurements from the operational radiosonde station at GAIA (see Table 5). A significant dry bias is found in the GAIA humidity observations, both with respect to the RS41 sondes (-2.9 $\mathrm{kg\,m^{-2}}$) and to the GNSS results (-1.2 $\mathrm{kg\,m^{-2}}$). The RS41 sondes and GNSS measurements show also a systematic difference, with a mean of 1.64 $\mathrm{kg\,m^{-2}}$ or 5 %, where the GNSS retrievals are drier than the sonde measurements. The IWV estimates from a colocated MWR agree with the RS41 results after an instrumental update on 27 January, while they exhibit a dry bias compared to GNSS and sondes before that date.

Understanding the origins of the various moisture biases requires further investigation that is beyond the scope of this paper. However, the GNSS minus sonde bias resembles strongly the results reported by Ciesielski et al. (2014b) where GNSS IWV data were compared to corrected Vaisala RS92 radiosonde measurements at several sites in the Indian Ocean. These authors found a similar dry bias in the GNSS data although good agreement was found with MWR measurements. However, the same authors also found systematic dry biases in satellite microwave data and in NWP model analyses/reanalyses compared to their

corrected radiosonde data. Other studies found either dry or wet biases, or no bias at all, between GNSS, radiosondes, and other techniques (see e.g. Buehler et al. (2012) and references therein, and also Bock et al. (2007); Wang and Zhang (2008); Yoneyama et al. (2008); Ning et al. (2012)).

A global dry bias intrinsic in the GNSS technique is unlikely. Instead, several error sources might contribute with variable signs and magnitudes such that the resulting bias appears strongly station-dependent. It has been observed that at sites with

multipath, satellite visibility obstructions, and/or electromagnetic interferences, the ZTD estimates can be biased either dry or wet (Ning et al., 2011). Biases can also result from using wrong antenna models or inaccurate mapping functions. Such situations can be detected by performing cutoff tests and can be partly mitigated by using a higher cutoff angle. We checked the BCO GNSS data by reprocessing the measurements using two different cutoff angles, one lower (5°) and one higher (10°) than the nominal value (7°) and found negligible impact on the mean ZTD estimates. This result excludes the hypothesis of any

of those bias sources. However, GNSS is not an absolute remote sensing technique and unless the IWV estimates are compared to an adequate reference it is difficult to figure out which part of the observed bias is due to GNSS or to the RS41 sondes.

IWV estimates from the ERA5 reanalysis were also compared to GNSS data and to BCO and GAIA sonde data. It was found that the IWV content from reanalysis over Barbados is overall close to the GAIA observations, probably due to the assimilation of the GAIA sondes in the reanalysis. However, at several occasions, ERA5 is shown to significantly underestimate IWV peaks

observed by all systems (sondes, GNSS, and MWR) by 5 to 8 $\mathrm{kg\,m^{-2}}$ (see Fig. 5). Two such events are documented (22 January and 23/24 January) during which a deep moist layer extended from the surface up to altitudes of 3.5 and 5 km (see Fig. 7). It was shown that ERA5 significantly underestimated the moisture content in the upper part of these layers, possibly due to the assimilation of other data over the domain that were biases low. Overall, the reanalysis showed a small dry bias (0.3 $\mathrm{kg\,m^{-2}}$) over the study area in comparison to the 49 GNSS stations. The origin of the ERA5 mean and occasional biases

needs further investigation.

At the synoptic scale, ERA5 showed spatio-spatial variations in the IWV field over the domain which were in general good agreement with the observations from the GNSS network. The link with the cloud organisation was studied using MODIS visible images inspired by the classification of Stevens et al. (2020). We found that the environment of Fish cloud patterns was moister than that of Flowers cloud patterns which, in turn, is moister than the environment of Gravel cloud patterns.

The differences in the IWV means between Fish and Gravel were assessed to be significant. Finally, the Gravel moisture environment was found to be similar to that of clear, cloud-free conditions. The moisture environment associated with the Sugar cloud pattern has not been assessed because it was hardly observed during the first two months of 2020. These preliminary results prompt for a more systematic analysis of the cloud organisation and the lower and mid-tropospheric moisture field.

*Data availability.* The GNSS RINEX data used in this work are available from the following data servers:

– RGP: ftp://rgpdata.ign.fr/pub/data/, last access : 29 January 2021;

    – UNAVCO: ftp://data-out.unavco.org/pub/rinex/, last access : 29 January 2021;

    – ORPHEON: http://reseau-orpheon.fr/, last acess : 29 January 2021;

    – SONEL: https://www.sonel.org/, last access : 29 January 2021.

The ORPHEON GNSS RINEX data are provided for scientific use in the framework of the GEODATA-INSU-CNRS convention. The DOI
references for the UNAVCO GNSS RINEX data are listed in Table A1 in the Appendix. The GIPSY reprocessed ZTD and IWV estimates from the 49 GNSS stations used in this study are available from AERIS, the French national data and service portal for the atmosphere (https://www.aeris-data.fr, last access : 29 January 2021), under DOI https://doi.org/10.25326/79 (Bock, 2020b).

https://doi.org/10.5194/jn-0-1-2021-supplement

*Author contributions.* OB prepared the GNSS equipment for stations BCON and BCOS. ED installed the stations at the BCO. FJ provide
technical support with equipment shipping, installation, Internet access, and routine functioning. RF and PB did the GNSS data processing with Bernese and GIPSY OASIS II software, respectively. FJ provided the BCO radiosonde data. SS provided the HATPRO IWV data. SB organized the EUREC[4]A campaign. OB, PB, and CF did the analysis and co-wrote the manuscript. All the authors participated in the discussion of the results.

*Competing interests.* The authors declare that they have no conflict of interests.

Special issue statement. This article is part of the special issue "Elucidating the role of clouds–circulation coupling in climate: datasets from the 2020 (EUREC[4]A) field campaign". It is not associated with a conference.

Financial support. This work was supported by the CNRS program LEFE/INSU through the project VEGAN. The EUREC[4]A project was supported by the European Research Council (ERC) under the European Union's Horizon 2020 research and innovation programme (grant agreement no. 694768).



*Acknowledgements.* We acknowledge the GNSS data providers in the Caribbean region, namely:

- RESIF, a French national Research Infrastructure, recognised as such by the French Ministry of Higher Education and Research. RESIF is managed by the RESIF Consortium, composed of 18 Research Institutions and Universities in France. RESIF is additionally supported by a public grant overseen by the French National Research Agency (ANR) as part of the "Investissements d'Avenir" program (reference: ANR-11-EQPX-0040) and the French Ministry of Ecology, Sustainable Development and Energy.

- ORPHEON, the French Real Time Kinematic service of Géodata Diffusion, part of Hexagon since 2014.

- GAGE Facility operated by UNAVCO, Inc., with support from the National Science Foundation and the National Aeronautics and Space Administration under NSF Cooperative Agreement EAR-1724794.

We acknowledge the use of imagery from the NASA Worldview application (https://worldview.earthdata.nasa.gov/), part of the NASA Earth Observing System Data and Information System (EOSDIS). We thank AERIS, the French data and service centre for atmosphere, for 675 providing the ERA5 reanalysis data and hosting the GNSS data.





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



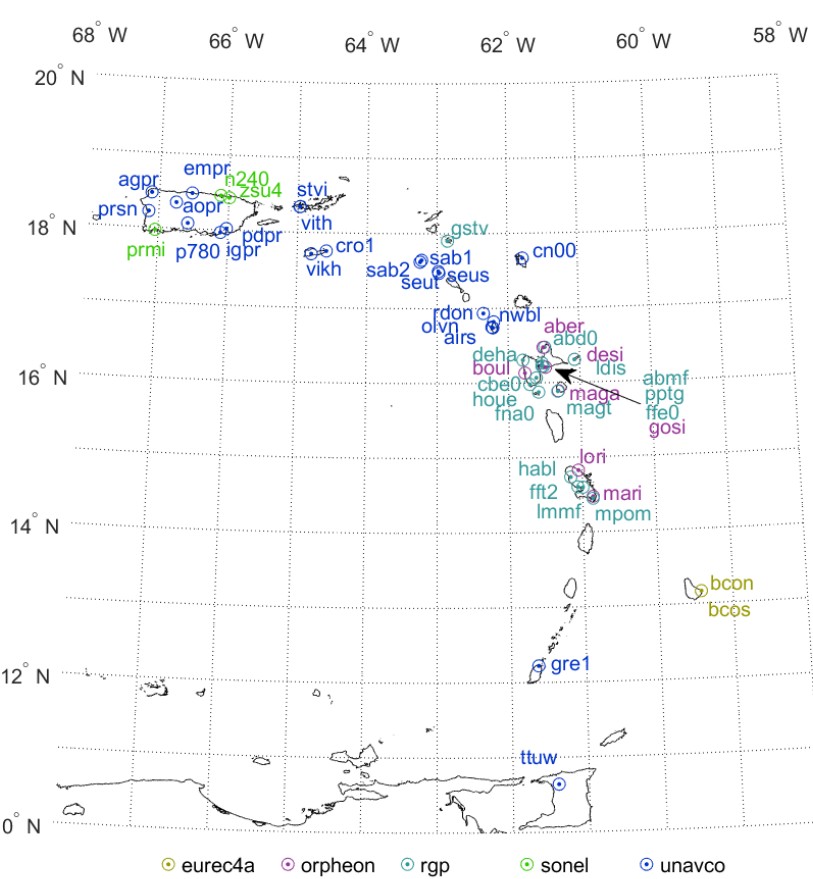

**Figure 1.** Map of GNSS stations for which the measurements were reprocessed from 1 January to 29 February 2020. The data sources are indicated below the figure.



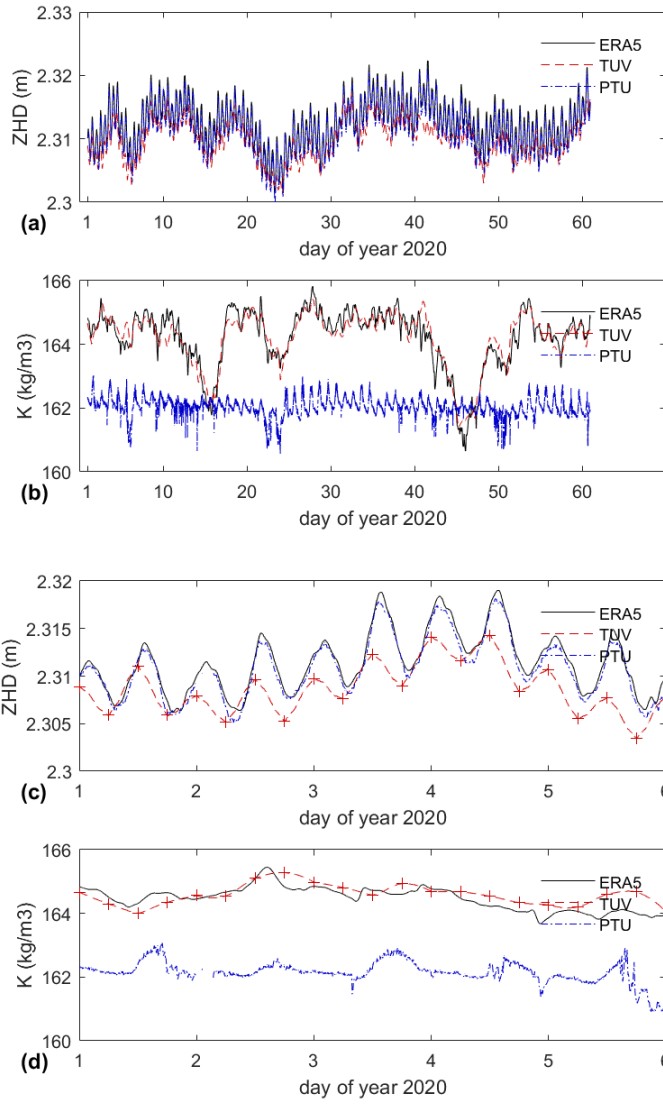

**Figure 2.** Time series of the auxiliary data ($ZHD$ and $K$) used to convert ZTD to IWV in the different GNSS streams: high resolution ERA5 pressure level data, ECMWF analysis provided by TUV, surface pressure and temperature observations (PTU) for GNSS station BCON. Top: from 1 January to 29 February 2020. Bottom: from 1 to 6 January 2020.



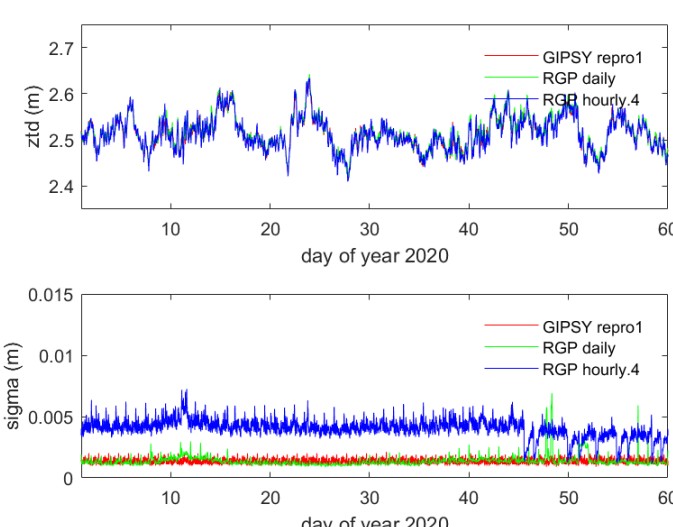

**Figure 3.** Time series of ZTD estimates and formal errors (sigma) from RGP NRT.4, RGP daily, and GIPSY repro1 solutions from 1 January to 29 February 2020.

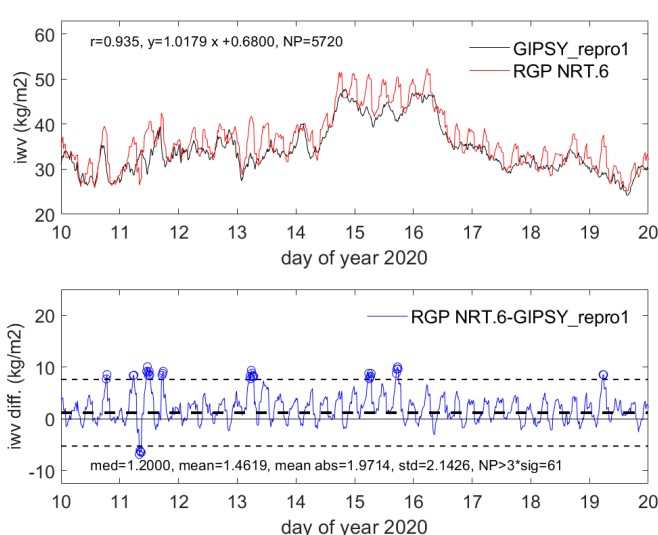

**Figure 4.** Time series of IWV from near real time GNSS processing (RGP NRT.6) compared to post-processed GNSS (GIPSY repro1) from 10 to 20 January 2020. The upper plot shows the IWV time series and the lower plot the IWV difference (NRT minus repro1).

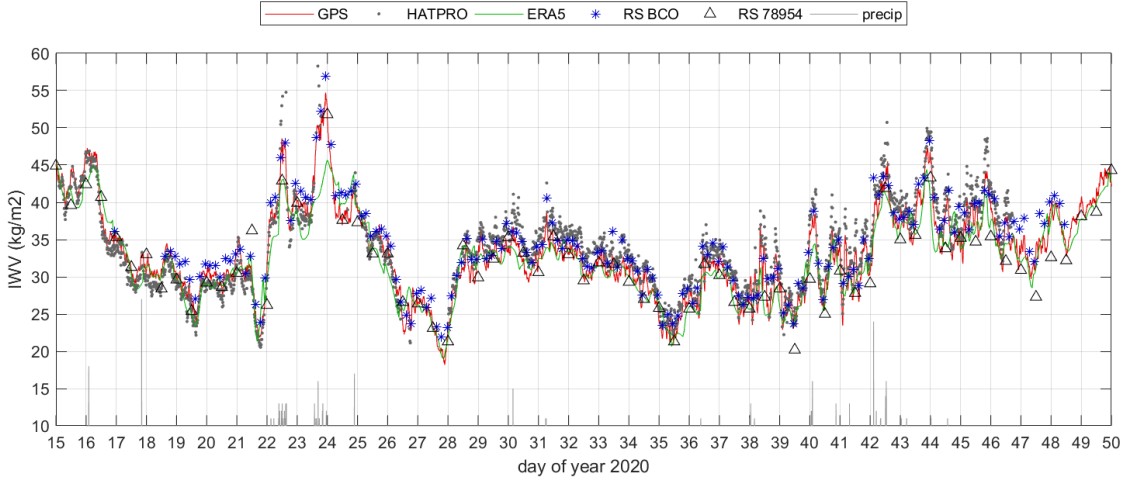

**Figure 5.** Time series of IWV from GNSS station BCON, RS41 sondes and HATPRO microwave radiometer colocated at BCO, as well as operational sondes from Grantley Adams International Airport (WMO ID 78954) and ERA5 reanalysis, from 15 January to 19 February 2020. The grey vertical bars at the bottom of the plot show precipitation data collected at BCO.



**Figure 6.** Pairwise scatter diagrams of IWV estimates from GNSS station BCON (GPS), HATPRO MWR, Vaisala RS41 radiosondes released from the BCO (RS BCO), operational radiosondes released from Grantley Adams International Airport (RS GA), and ERA5 reanalysis. The HATPRO data set has been separated in two parts, before (black symbols) and after (red) the 27 January when the system was fixed for an instrumental failure. Note that sample sizes are different between diagrams because each comparison is done with the highest temporal resolution for the period of available data between 1 January and 29 February 2020.



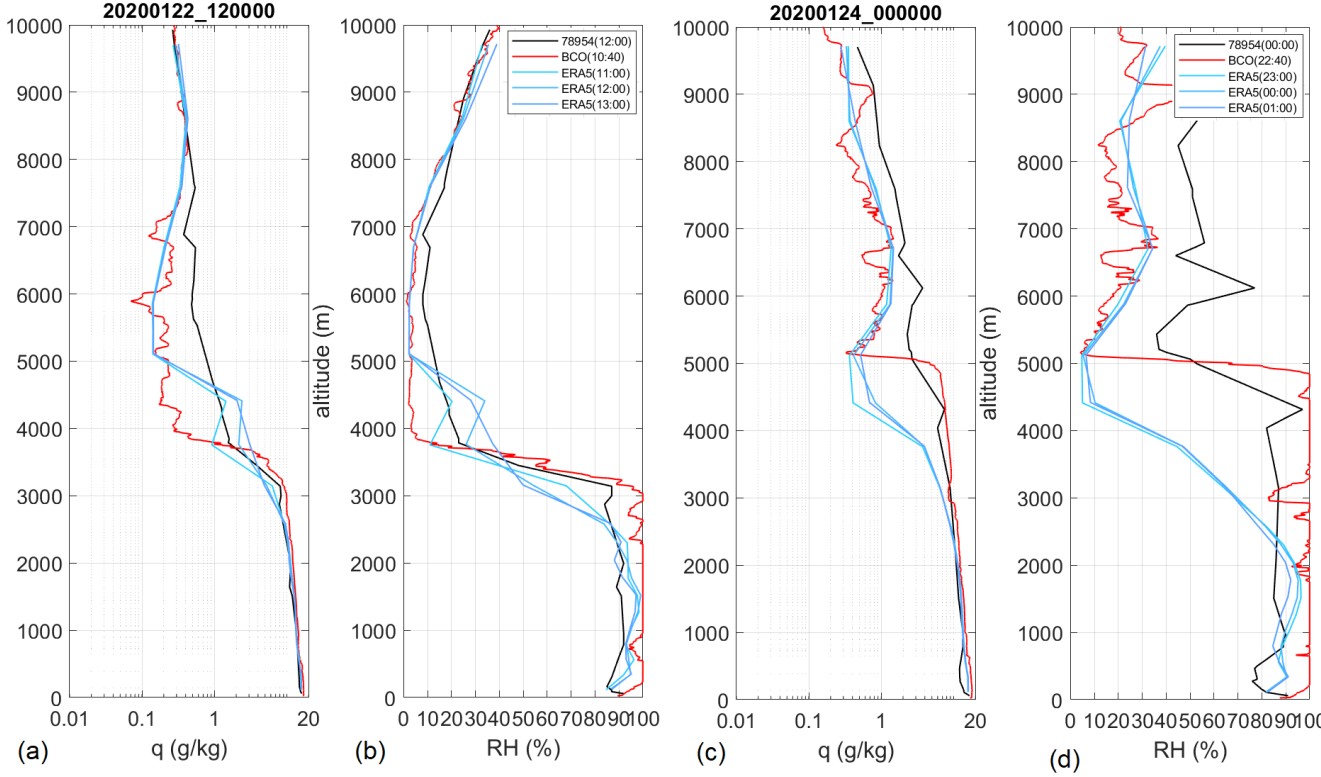

**Figure 7.** Humidity profiles from BCO and Grantley Adams radiosondes(WMO ID 78954) on (a, b) 22 January 2020 at 12 UTC and (c, d) 24 January 2020 at 00 UTC (exact times are indicated in the plot legend).

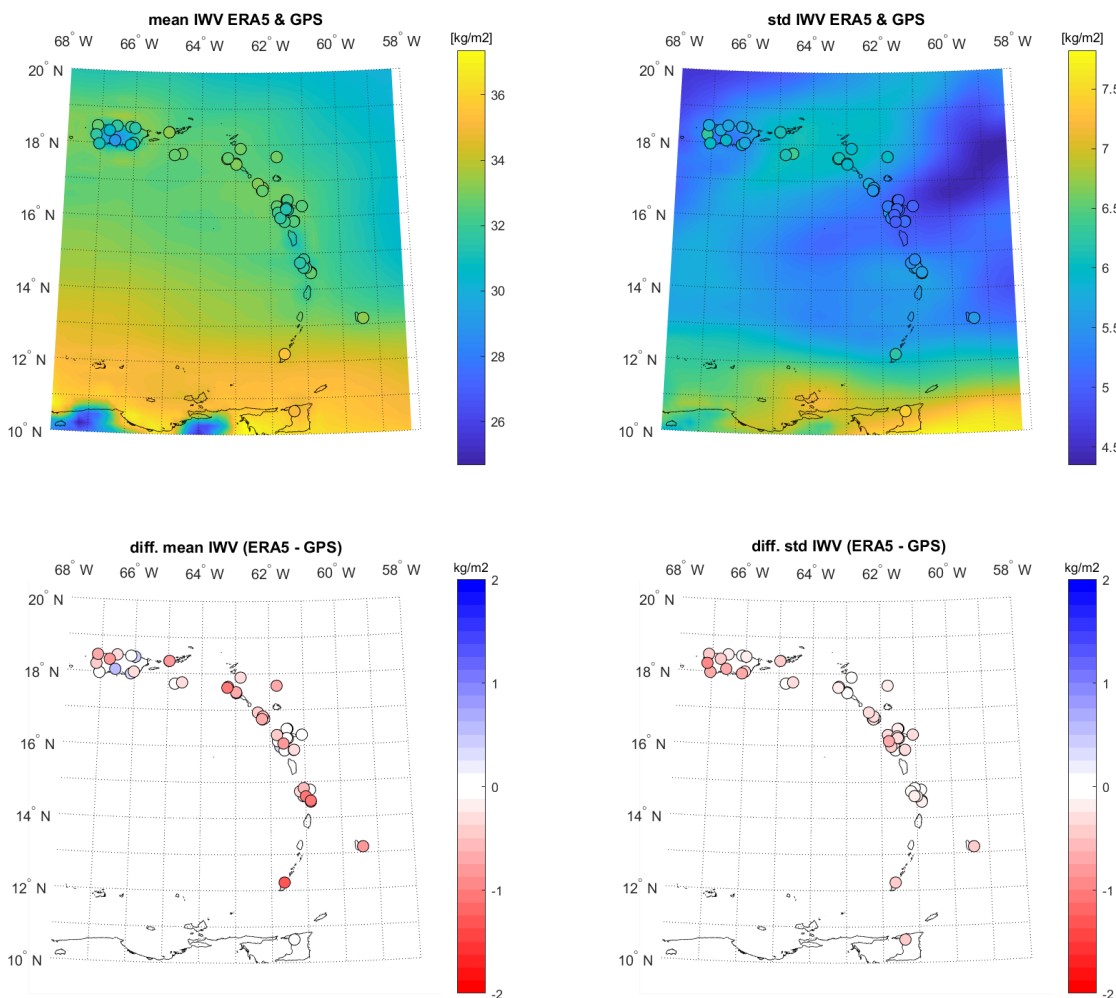

**Figure 8.** Upper plots: mean and standard deviation of hourly IWV from ERA5 (background) and GNSS (circles) from 1 January to 29 February 2020. Lower plots: difference of mean and standard deviation of hourly IWV values (ERA5 - GNSS). In the upper plots, the GNSS IWV data have been height corrected and gaps have been filled with ERA5 values to minimize representativeness differences. In the lower plots, ERA5 IWV data have computed from pressure level profiles from the height of the GNSS stations upwards (no correction applied to GNSS IWV data).



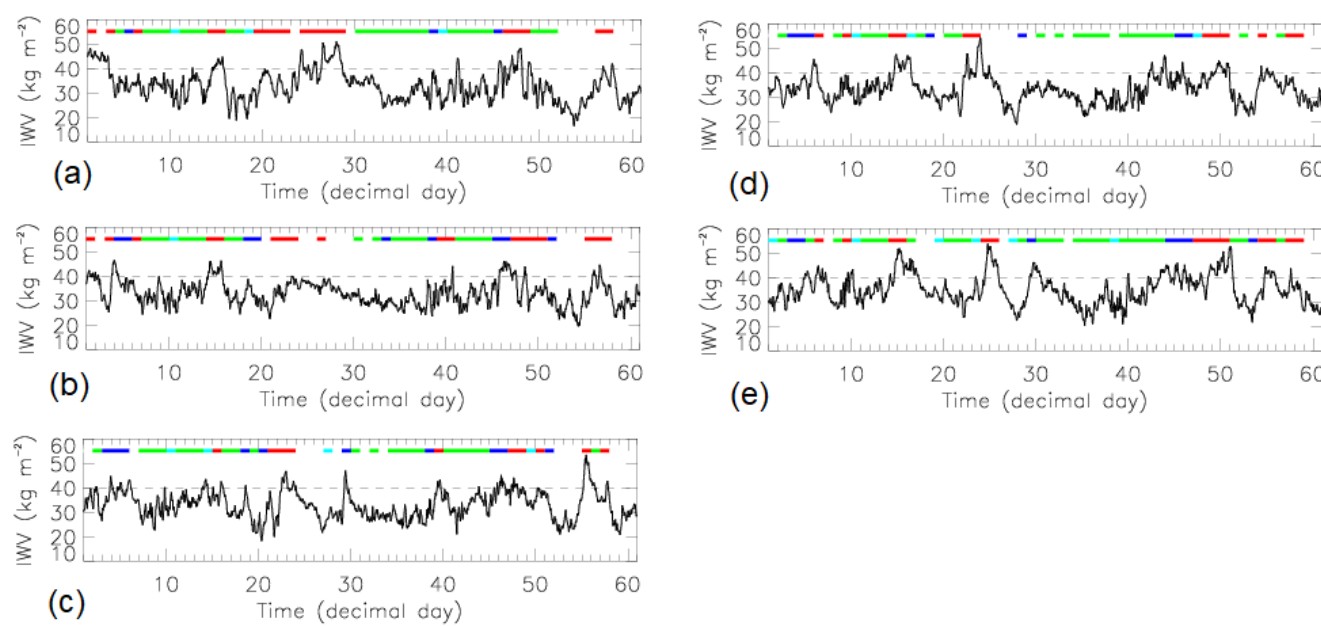

**Figure 9.** Time series of IWV from five GNSS stations for the period from 1 January to 29 February 2020 as well as the type of cloud patterns in the vicinity of the GNSS sites (see color code in Fig. 11).



**Figure 10.** Visible image from (upper left) MODIS Aqua and (upper right) Terra, on 19 January 2020, and (lower) MODIS Aqua, on 13 January 2020, depicting different cloud organisations over the study domain (57.4-67.6° W and 8.9-19.1° N). The yellow arrows locate St Croix (north-west) and Barbados (south-east) in the upper left image, and Guadeloupe, Martinique and Grenade (from north to south) in the upper right image. In the lower image, a Gravel cloud organisation is observed uniformly across the domain on 13 January 2020. Source: NASA WorldView (https://worldview.earthdata.nasa.gov/, last access: 29 January 2021).


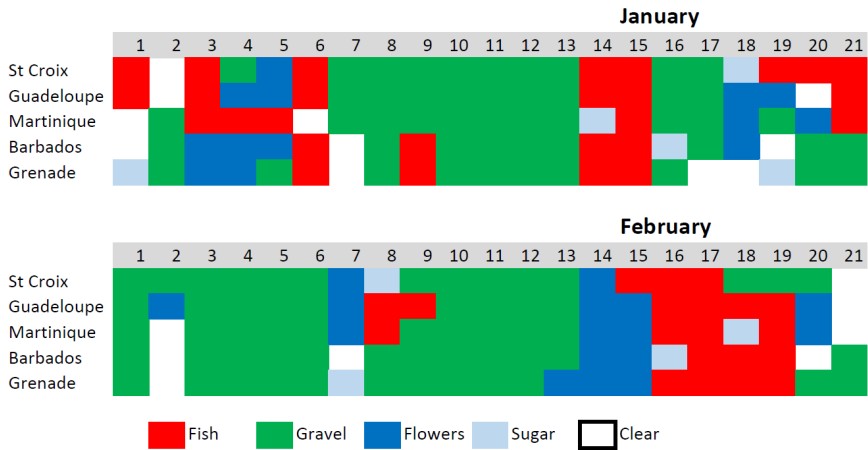

**Figure 11.** Cloud type classification performed for each of the five sites in January and February 2020.

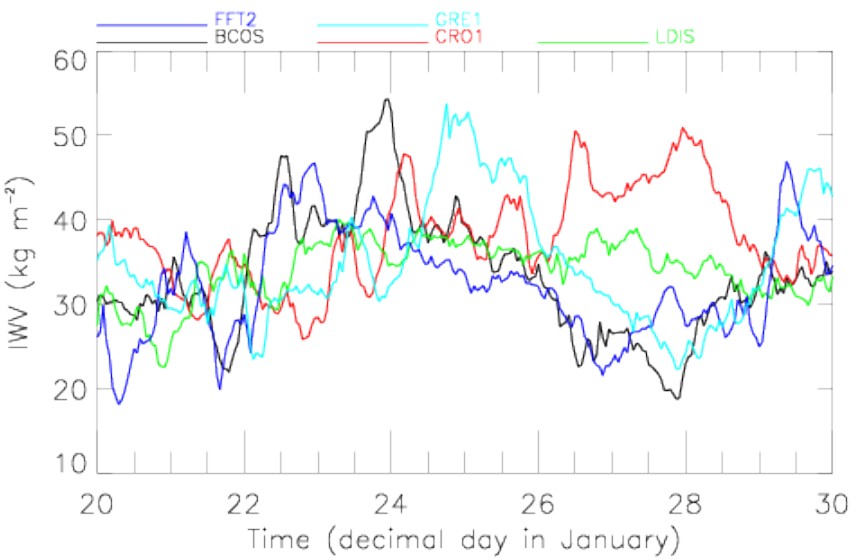

**Figure 12.** IWV time series observed from GNSS over each of the five sites between 20 and 30 January 2020.

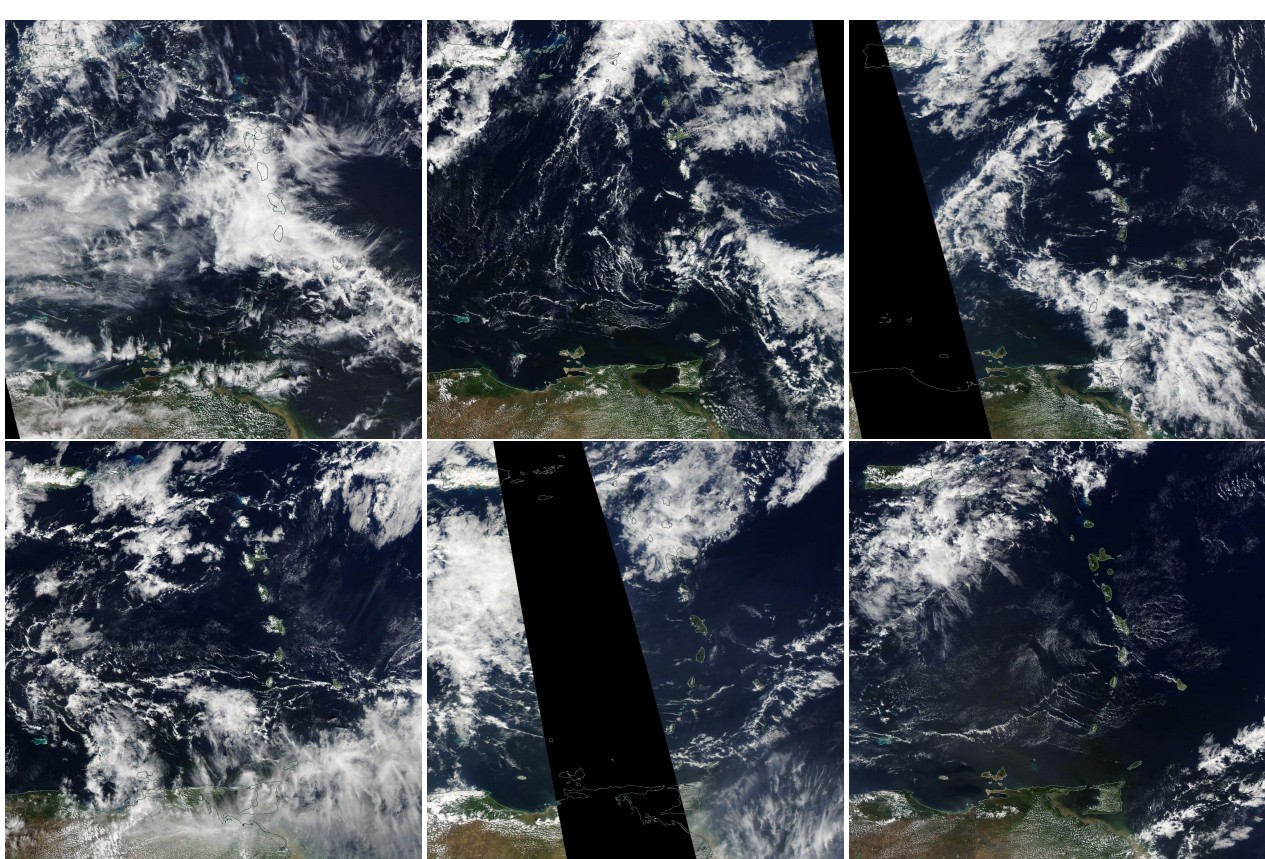

**Figure 13.** Visible images from MODIS Aqua around 1330 UTC on each day between 22 and 27 January 2020 (from left to right and top to bottom). Source: NASA WorldView (https://worldview.earthdata.nasa.gov/, last access: 29 January 2021).

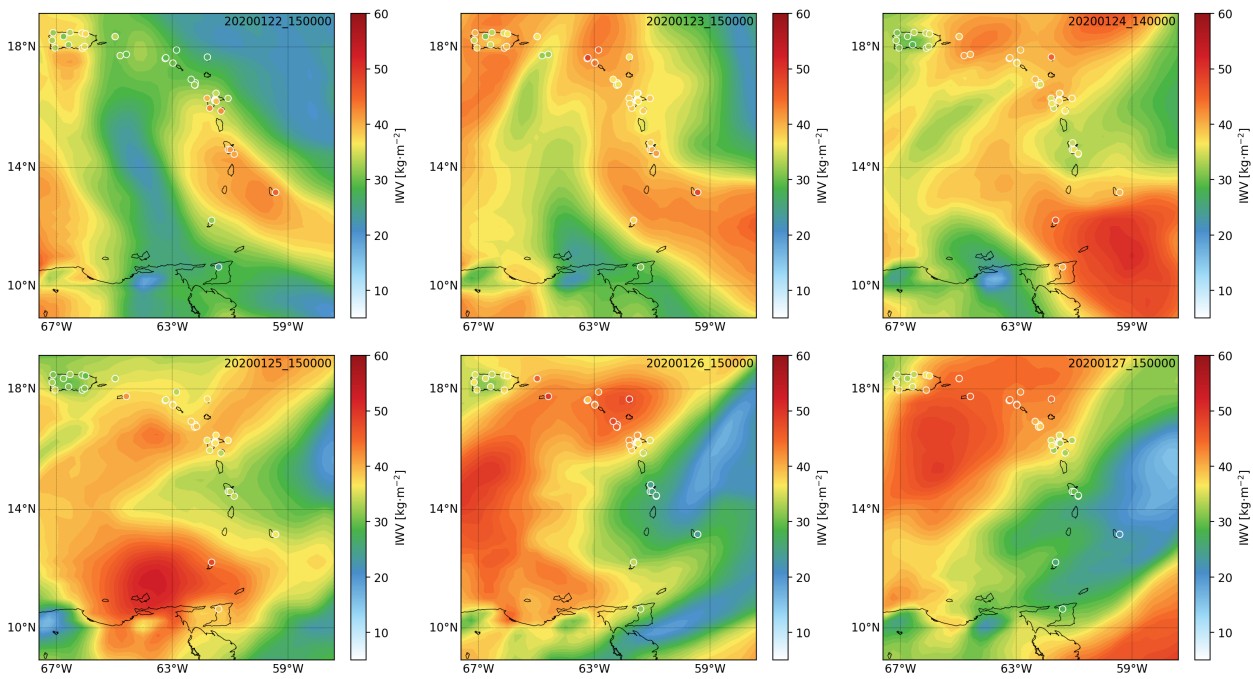

**Figure 14.** IWV fields from ERA5 reanalysis (background) and GNSS (circles) at 1500 UTC on each day between 22 and 27 January 2020 (see text insert in each image).





**Table 1.** Details of the five processing streams operated by IGN and ENSTA_B/IPGP in the framework of EUREC[4]A.

| **RGP NRT** | |
|---|---|
| Availability | 31 October 2019 - present |
| Software | Bernese GNSS software v5.2 |
| Strategy | Double-difference solution |
| Network | 30 to 40 permanent stations from South America to Canada |
| Orbits and clocks | Ultra-rapid IGS |
| Window length | 6 h, shifted by 1 h every hour |
| Elevation cutoff angle | 10° |
| Observation sampling | 30 s |
| Observation weighting | $\sigma^2 = 10^{-6}/\cos^2(Z)$ where Z = zenith angle |
| Tropospheric model | ZHD and ZWD a priori : GPT model |
| | ZHD and ZWD mapping functions : GMF ; Gradient mapping function : tilting |
| | ZWD sampling : 15 min ; Gradient sampling : 1 per window |
| | ZWD constraint : 5 m absolute and 1 mm relative ; gradient constraint : 5 m |
| Coordinate estimates | Fixed |
| Ambiguity resolution | quasi-ionosphere free (baselines ≥ 20 km) and Sigma (baselines < 20 km) |
| Specific features | Re-use of the normal equation from previous solution |
| (NRT only) | |

| **RGP daily** | (difference wrt RGP NRT) |
|---|---|
| Availability | 31 October 2019 - present |
| Network | 70 to 80 permanent stations from South America to Canada |
| Orbits and clocks | Rapid ESA (European Space Agency) |
| Window length | 24 h |
| Tropospheric model | ZWD constraint : 5 m absolute and relative |

| **GIPSY rapid** | |
|---|---|
| Availability | 31 October 2019 - present |
| Software | GIPSY OASIS II software v6.4 |
| Strategy | Precise Point Positioning (PPP) solution |
| Orbits and clocks | Rapid JPL (Jet Propulsion Laboratory) |
| Window length | 30 h (21UTC day D-1 to 03UTC day D+1) |
| Elevation cutoff angle | 7° |
| Observation sampling | 5 min |
| Observation weighting | Uniform weighting |
| Tropospheric model | ZHD and ZWD a priori : 6-hourly ECMWF analysis (provided by TUV) |
| | ZHD and ZWD mapping functions : VMF1 ; Gradient mapping function : Bar Sever, 1998 |
| | ZWD and gradient model : Random Walk |
| | ZWD and gradient sampling : 5 min |
| | ZWD and gradient constraints : 5 mm h$^{-1/2}$ (ZWD) and 0.5 mm h$^{-1/2}$ (gradients) |
| Coordinate estimates | Estimated once per window |
| Ambiguity resolution | Bertiger et al., 2010 |

| **GIPSY final** | (difference wrt GIPSY rapid) |
|---|---|
| Orbits and clocks | Final JPL |

| **GIPSY repro1** | (difference wrt GIPSY rapid) |
|---|---|
| Availability | 1 January – 29 February 2020 |
| Orbits and clocks | Final JPL |
| Ionospheric model | Order 2 |



**Table 2.** RGP NRT ZTD and IWV solutions (NRT.1 to NRT.6) compared to RGP daily and GIPSY repro1, station BCON, period 1 January 2020 to 29 February 2020. In the upper row, 'Sigma' indicates the mean formal error of the RGP NRT solutions (in meters). The comparisons show: mean differences (RGP NRT minus RGP daily/GIPSY repro1), standard deviation of differences and the number of points (NP). The RGP daily data sampling is 1 hour and the GIPSY repro1 sampling is 5 min.

| ZTD | NRT.1 | NRT.2 | NRT.3 | NRT.4 | NRT.5 | NRT.6 |
|---|---|---|---|---|---|---|
| Sigma (m) | 0.0034 | 0.0034 | 0.0036 | 0.0040 | 0.0046 | 0.0061 |
| Compared to RGP daily (1-hourly) | | | | | | |
| Mean diff. (m) | -0.0406 | -0.0314 | -0.0190 | -0.0076 | 0.0018 | 0.0076 |
| Std. Diff. (m) | 0.0169 | 0.0127 | 0.0103 | 0.0094 | 0.0104 | 0.0129 |
| NP | 1436 | 1437 | 1438 | 1439 | 1440 | 1438 |
| Compared to GIPSY repro1 (5-min) | | | | | | |
| Mean diff. (m) | -0.0360 | -0.0243 | -0.0123 | -0.0016 | 0.0066 | 0.0108 |
| Std. Diff. (m) | 0.0154 | 0.0126 | 0.0107 | 0.0102 | 0.0111 | 0.0131 |
| NP | 5733 | 5737 | 5741 | 5745 | 5743 | 5724 |
| IWV | NRT.1 | NRT.2 | NRT.3 | NRT.4 | NRT.5 | NRT.6 |
| Compared to RGP daily (1-hourly) | | | | | | |
| Mean diff. (kg m$^{-2}$) | -6.58 | -5.10 | -3.08 | -1.24 | 0.29 | 1.23 |
| Std. Diff. (kg m$^{-2}$) | 2.73 | 2.06 | 1.67 | 1.52 | 1.69 | 2.08 |
| NP | 1434 | 1435 | 1436 | 1437 | 1438 | 1437 |
| Compared to GIPSY repro1 (5-min) | | | | | | |
| Mean diff. (kg m$^{-2}$) | -6.13 | -4.24 | -2.29 | -0.55 | 0.77 | 1.46 |
| Std. Diff. (kg m$^{-2}$) | 2.48 | 2.03 | 1.74 | 1.66 | 1.82 | 2.14 |
| NP | 5727 | 5731 | 5735 | 5739 | 5738 | 5720 |





**Table 3.** Similar to Table 2, but for RGP daily, GIPSY rapid and GIPSY repro1 results, for stations BCON and BCOS. The leftmost part shows ZTD results and the rightmost shows IWV results.

| ZTD | BCON | BCOS | IWV | BCON | BCOS |
|---|---|---|---|---|---|
| RGP_daily.vs.GIPSY_rapid | | | | | |
| Mean diff. (m) | 0.0021 | 0.0011 | Mean diff. (kg m$^{-2}$) | -0.390 | -0.550 |
| Std. Diff. (m) | 0.0058 | 0.0057 | Std. Diff. (kg m$^{-2}$) | 1.000 | 1.000 |
| NP | 1440 | 1440 | NP | 1438 | 1438 |
| RGP_daily.vs.GIPSY_repro1 | | | | | |
| Mean diff. (m) | 0.0024 | 0.0013 | Mean diff. (kg m$^{-2}$) | 0.085 | -0.085 |
| Std. Diff. (m) | 0.0058 | 0.0057 | Std. Diff. (kg m$^{-2}$) | 0.955 | 0.942 |
| NP | 1440 | 1440 | NP | 1438 | 1438 |
| GIPSY_rapid.vs.GIPSY_final | | | | | |
| Mean diff. (m) | 0.0003 | 0.0003 | Mean diff. (kg m$^{-2}$) | 0.055 | 0.048 |
| Std. Diff. (m) | 0.0008 | 0.0008 | Std. Diff. (kg m$^{-2}$) | 0.129 | 0.129 |
| NP | 17275 | 17280 | NP | 17275 | 17280 |
| GIPSY_rapid.vs.GIPSY_repro1 | | | | | |
| Mean diff. (m) | 0.0003 | 0.0002 | Mean diff. (kg m$^{-2}$) | 0.473 | 0.463 |
| Std. Diff. (m) | 0.0018 | 0.0017 | Std. Diff. (kg m$^{-2}$) | 0.468 | 0.480 |
| NP | 17268 | 17272 | NP | 17266 | 17264 |
| GIPSY_final.vs.GIPSY_repro1 | | | | | |
| Mean diff. (m) | -0.0001 | -0.0001 | Mean diff. (kg m$^{-2}$) | 0.418 | 0.415 |
| Std. Diff. (m) | 0.0016 | 0.0016 | Std. Diff. (kg m$^{-2}$) | 0.450 | 0.461 |
| NP | 17266 | 17272 | NP | 17266 | 17264 |





**Table 4.** Separation of bias (mean) and random errors (one standard deviation) of IWV differences into contributions from ZTD differences, ZHD differences, and K differences, for two operational data streams (RGP daily and GIPSY rapid) and the reprocessed stream (GIPSY repro1). The three streams involve different auxiliary data for the ZTD to IWV conversion from PTU, TUV, and ERA5. See text for more details.

| ZTD data | Auxiliary data | Contribution to bias (kg m$^{-2}$) | | | | Contribution to random error (kg m$^{-2}$) | | | |
|---|---|---|---|---|---|---|---|---|---|
| | | ZTD | ZHD | K | total | ZTD | ZHD | K | total |
| RGP_daily vs GIPSY_rapid | PTU vs TUV | 0.34 | 0.29 | -0.44 | -0.39 | 0.92 | 0.36 | 0.17 | 0.99 |
| RGP_daily vs GIPSY_repro1 | PTU vs ERA5 | 0.39 | -0.13 | -0.44 | 0.08 | 0.94 | 0.08 | 0.18 | 0.95 |
| GIPSY_rapid vs GIPSY_repro1 | TUV vs ERA5 | -0.02 | -0.43 | 0.00 | 0.41 | 0.28 | 0.36 | 0.06 | 0.46 |



**Table 5.** Pairwise comparison of IWV estimates from GNSS station BCON (GPS), HATPRO MWR, Vaisala RS41 radiosondes released from the BCO (RS BCO), operational radiosondes released from Grantley Adams International Airport (RS GA), and ERA5 reanalysis. The HATPRO data set has been separated in two parts, before (back symbols) and after (red) the 27 January when the system was fixed for an instrumental failure. For each period the data from all sources have been time-matched as reflected from the number of data points (NP).

| Comparison | Mean diff. (kg m$^{-2}$) | Std. Diff. (kg m$^{-2}$) | Slope | Offset (kg m$^{-2}$) | R | NP |
|---|---|---|---|---|---|---|
| **Period 1 January – 29 February 2020** | | | | | | |
| BCO.vs.GPS | 1.6 | 0.85 | 1.06 | -0.21 | 0.99 | 60 |
| GA.vs.GPS | -1.26 | 1.59 | 0.99 | -1.17 | 0.96 | 60 |
| ERA5.vs.GPS | -1.02 | 1.64 | 0.93 | 1.27 | 0.96 | 60 |
| GA.vs.BCO | -2.86 | 1.84 | 0.94 | -0.65 | 0.95 | 60 |
| ERA5.vs.BCO | -2.62 | 1.91 | 0.92 | 0.37 | 0.96 | 60 |
| ERA5.vs.GA | 0.24 | 1.57 | 0.98 | 1.02 | 0.96 | 60 |
| **Period 1 January – 27 January 2020** | | | | | | |
| HATPRO.vs.GPS | -1 | 1.76 | 1.23 | -8.81 | 0.98 | 14 |
| HATPRO.vs.BCO | -2.36 | 1.15 | 1.11 | -6.18 | 0.99 | 14 |
| HATPRO.vs.GA | -0.16 | 2.59 | 1.16 | -5.38 | 0.92 | 14 |
| HATPRO.vs.ERA5 | -0.13 | 2.09 | 1.18 | -5.91 | 0.96 | 14 |
| **Period 28 January – 15 February 2020** | | | | | | |
| HATPRO.vs.GPS | 2.08 | 0.83 | 1.13 | -1.96 | 0.99 | 36 |
| HATPRO.vs.BCO | 0.53 | 0.77 | 1.06 | -1.5 | 0.99 | 36 |
| HATPRO.vs.GA | 3.26 | 1.69 | 1.11 | -0.28 | 0.95 | 36 |
| HATPRO.vs.ERA5 | 2.97 | 1.65 | 1.14 | -1.74 | 0.96 | 36 |





**Table 6.** Upper part: number of cases over the five sites of interest in the Caribbean Arc in January and February 2020 for each cloud organisation type, Fish, Flower, Gravel and Sugar, or lack thereof, according to the classification of (Stevens et al., 2020). The cases were identified by visual inspection of the MODIS Aqua and Terra visible images at 1330 and 1030 local equator crossing time available from NASA WorldView (https://worldview.earthdata.nasa.gov/, last access: 29 January 2021). Lower part: mean and standard deviation of GNSS-derived IWVs associated with Fish, Gravel, Flowers and clear conditions.

| Station | St Croix (CRO1) | | Guadeloupe (LDIS) | | Martinique (FFT2) | | Barbados (BCOS) | | Grenade (GRE1) | |
|---|---|---|---|---|---|---|---|---|---|---|
| **Cloud organisation type** | | | | | | | | | | |
| Fish | 19 | | 19 | | 10 | | 13 | | 14 | |
| Flower | 3 | | 10 | | 11 | | 8 | | 8 | |
| Gravel | 26 | | 19 | | 23 | | 23 | | 27 | |
| Sugar | 3 | | 0 | | 3 | | 2 | | 6 | |
| Clear | 10 | | 12 | | 13 | | 14 | | 5 | |
| **IWV mean and standard deviation (kg m$^{-2}$)** | | | | | | | | | | |
| | Mean | SD | Mean | SD | Mean | SD | Mean | SD | Mean | SD |
| Fish | 38.1 | 4.1 | 35.1 | 3.5 | 38.3 | 4 | 37.6 | 4.8 | 40.5 | 4.3 |
| Flower | 30.9 | 2.9 | 30.8 | 2.1 | 31.4 | 3.5 | 31.5 | 3.7 | 33.1 | 4.6 |
| Gravel | 33.3 | 1.9 | 34.9 | 2.9 | 35.1 | 5.1 | 34.4 | 5 | 35.6 | 6.7 |
| Clear | 29.7 | 6.7 | 30.1 | 4.4 | 32.1 | 7.3 | 32.4 | 6 | 33.2 | 5.2 |





**Table 7.** Significance of differences in the means of IWV samples using a Student's t-test across all stations. 'Yes' means that the means are statistically different at the level 0.05.

|         | Fish | Gravel | Flowers | Clear |
|---------|------|--------|---------|-------|
| Fish    |      | Yes    | No      | Yes   |
| Gravel  | Yes  |        | No      | No    |
| Flowers | No   | No     |         | No    |
| Clear   | Yes  | No     | No      |       |





## Appendix A: Updated refractivity coefficients and their uncertainty

The calculation of $ZHD$ and $\kappa(Tm)$ from Eq. (2) and (4) involves refractivity coefficients $k_1$, $k_2$, and $k_3$, and specific gas constants for dry air and water vapour ($R_d$ and $R_v$), with $k_2' = k_2 - k_1 \times (R_d/R_v)$. The specific gas constants that we used hereafter are $R_d = 287.001$ J K$^{-1}$ kg$^{-1}$ for dry air (which includes an updated mixing ratio of 408 ppm for $CO_2$) and $R_v = 461.522$ J K$^{-1}$ kg$^{-1}$ (Kestin et al., 1984).

Many authors published refractivity coefficients from experimental work performed between the 1950s and the 1970s. Smith
and Weintraub (1953) compiled and averaged the early measurements, and Hasegawa and Stokesberry (1975) compiled and characterized a significantly larger number of experimental results. Thayer (1974) developed an alternative and hybrid approach which includes measurements extrapolated from optical frequencies. Bevis et al. (1994) revisited the data used by Hasegawa and Stokesberry (1975) and determined a new set of average values and associated uncertainties. Finally, Rueger (2002) proposed a new set of „best average" coefficients after reassessing the data set used by Bevis et al. (1994). While there has been
a broad consensus on the value of $k_1$ among previous authors, Rueger's new $k_1$ coefficient is 0.115 % larger than previous values. The impact on ZHD would be an increase of about 2.6 mm at mean sea level (i.e. a bias in IWV of $-0.4$ kg m$^{-2}$). The impact is also significant on the determination of bending angles from GNSS radio-occultation measurements as discussed by Healy (2011). The latter author examined the origin of the increase in Rueger's $k_1$ and identified two obvious reasons: a numerical inconsistency in the value of 0 °C = 273 K instead of 273.15 K and neglecting $CO_2$ in the gas mixture composing
the dry air in many previous studies. Healy (2011) highlights that although Rueger's estimate of $k_1$ appears to be more robust and defendable than the previous values, it also has one significant caveat as it does not account for non-ideal gas effects. According to the significant work done by Rueger (2002) in re-assessing past measurements and re-evaluating the refractivity coefficients we believe that his results are the more accurate to date and will use them along with a correction for the non-ideal gas effects, as suggested by Healy (2011), and an update for present $CO_2$ content.

We start with Rueger (2002)'s 'best average' coefficients $k_1'$=77.6681 K hPa$^{-1}$, $k_2$=71.2952 K hPa$^{-1}$, $k_3$=375463 K$^2$ hPa$^{-1}$, $k_4$=133.4800 K hPa$^{-1}$, where $k_4$ is the refractivity constant for $CO_2$ and $k_1'$ it the refractivity constant for dry air without $CO_2$. Using a present day $CO_2$ mixing ratio of $r_c$=408 ppm, $k_1'$ and $k_4$ can be summed together to form $k_1$, the refractivity constant for dry air including $CO_2$, $k_1 = k_1' \cdot (1-r_c) + k_4 \cdot r_c$, to give $k_1 = 77.6909$ K hPa$^{-1}$. The last step is correcting for the non-ideal gas effects using the compressibility factor given by Owens (1967), i.e. 1/Zd = 1.000588 for dry air at 273.15 K and 1013.25
hPa, and 1/Zw = 1.000698 for water vapour at 293.15 K and a partial pressure of 13.33 hPa (the conditions of measurements of refractivity used by Rueger (2002)). Finally, the updated refractivity coefficients become:

$$k_1 = 77.6452 \pm 0.0094 \text{ K hPa}^{-1}$$

$$k_2 = 71.2 \pm 1.3 \text{ K hPa}^{-1}$$

$$k3 = (3.7520 \pm 0.0076) \cdot 10^5 \text{ K}^2 \text{ hPa}^{-1}$$

where we included the uncertainties evaluated by Rueger (2002). If we assume that these uncertainties are fair estimates of the true absolute accuracy of the coefficients, the uncertainty in the IWV estimates due to $k_1$, $k_2$, and $k_3$, would be: 0.04, 0.03,





and 0.06 $\mathrm{kg\,m^{-2}}$, respectively, under the average conditions of EUREC[4]A (IWV = 32 $\mathrm{kg\,m^{-2}}$). The systematic bias due to the uncertainty in the refractivity coefficients is thus extremely small. However, using the new estimates can make some systematic difference compared to those used in past studies.

Figure A1 compares the new refractivity coefficients to those from Thayer (1974) and Bevis et al. (1994), which are the two most widely used data sources in GNSS meteorology, and those from Rueger (2002) (without the compressibility factor correction and assuming a $CO_2$ mixing ratio of 375 ppm). It is seen that the $k_1$ from Rueger is about 0.09 K $\mathrm{hPa^{-1}}$ larger than Thayer (1974) and Bevis et al. (1994) and that the new value differs only by 0.04 K $\mathrm{hPa^{-1}}$ from the latter which represents a ZHD difference of 1.22 mm and a IWV difference of -0.19 $\mathrm{kg\,m^{-2}}$ (the negative sign is because a larger ZHD correction

decreases the IWV estimate). The new value for $k_2$ is in agreement with Bevis et al. (1994) and Rueger (2002), but this coefficient has anyway a small weight in the final IWV estimate. The new value for $k_3$ is in good agreement with Rueger's and to a lesser extent with Bevis et al. (1994) from which it differs by -0.2399 $\cdot 10^5$ $\mathrm{K^2\,hPa^{-1}}$ leading to a small fractional change in IWV of -0.63 %, i.e. 0.20 $\mathrm{kg\,m^{-2}}$ assuming IWV = 32 $\mathrm{kg\,m^{-2}}$.

In conclusion, based on the differences between published refractivity coefficients and their uncertainties, we consider that

the uncertainty in the absolute IWV values (i.e. the possible bias) retrieved from GNSS during EUREC[4]A due to these coefficients is at the level of $\pm$ 0.2 $\mathrm{kg\,m^{-2}}$.

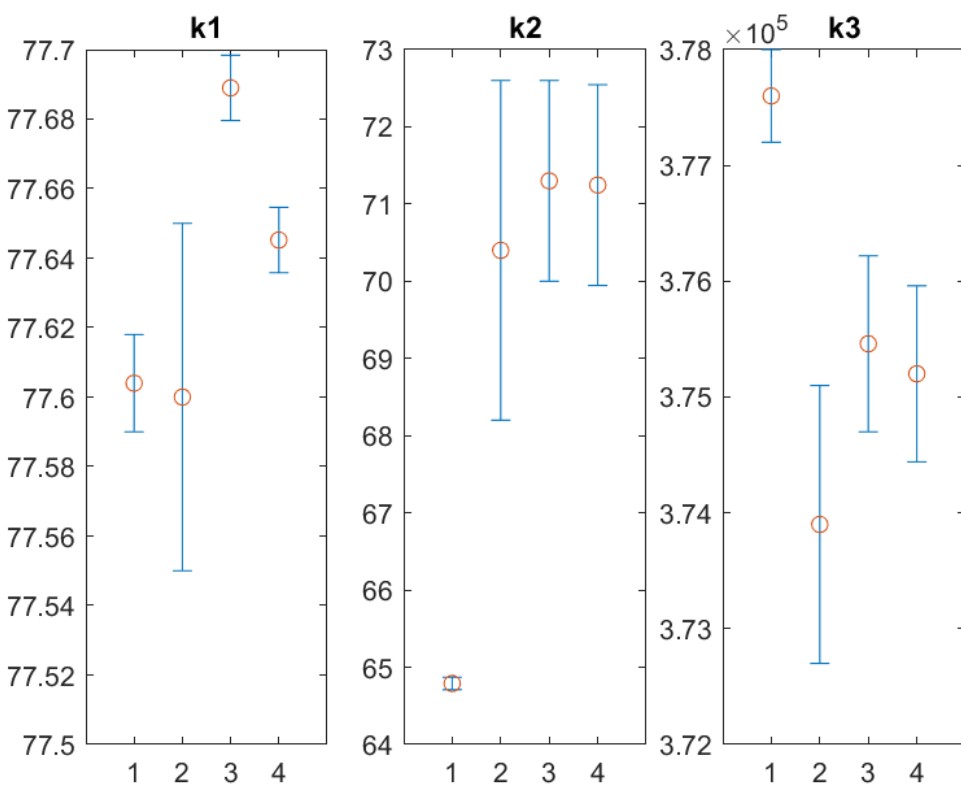

**Figure A1.** Comparison of refractivity coefficients from various authors: 1=Thayer (1974), 2=Bevis et al. (1994), 3=Rueger (2002), 4=this study.

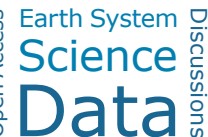

**Table A1.** List of Digital Object Identifiers associated with the Unavco GNSS stations from which RINEX data have been used in this study.

| Station ID | DOI |
| --- | --- |
| agpr | https://doi.org/10.7283/T5NC5ZZJ |
| airs | https://doi.org/10.7283/T53B5XGJ |
| aopr | https://doi.org/10.7283/T5HX1B1R |
| cn00 | https://doi.org/10.7283/T5FN14GQ |
| empr | https://doi.org/10.7283/T5930S05 |
| gre1 | https://doi.org/10.7283/T5BC3WZ5 |
| igpr | https://doi.org/10.7283/T5WW7GF4 |
| nwbl | https://doi.org/10.7283/T5ZK5F13 |
| olvn | https://doi.org/10.7283/T5Q23XMD |
| p780 | https://doi.org/10.7283/T54X55T3 |
| pdpr | https://doi.org/10.7283/T51N7ZX1 |
| prsn | https://doi.org/10.7283/T55D8QM9 |
| rdon | https://doi.org/10.7283/T5W37TFB |
| sab1 | https://doi.org/10.7283/633E-1497 |
| sab2 | https://doi.org/10.7283/TH2E-EQ61 |
| seus | https://doi.org/10.7283/RFYY-MM87 |
| seut | https://doi.org/10.7283/A49V-Z691 |
| stvi | https://doi.org/10.7283/T5QN653K |
| trnt | https://doi.org/10.7283/T5K935W2 |
| ttuw | https://doi.org/10.7283/T5TQ5ZTR |