# Peer review of "IWV observations in the Caribbean Arc from a network of ground-based GNSS receivers during EUREC4A"

_Earth System Science Data, 2021_

## Author Comment (AC1)

We would like to thank both reviewers for their detailed assessment of this manuscript and for their constructive comments. We have revised the manuscript accordingly. The reviewer's comments are repeated in black and our answers are given in blue. The changes made in the manuscript appear therein in red.

Please note that we updated the cloud pattern classification in Fig. 11 on two days: 14 and 15 February 2020. This change impacted marginally the statistics given in Table 6 but not the conclusions. Note also that Fig. 11 was cropped at day 21 in the initial submission and that Table 6 contained an inversion of line headers in the lower part of the Table (Flower and Gravel were inverted). Both glitches were fixed in the revised manuscript.

**Reviewer 1**

GENERAL

This is a nice paper on the quality of ZTD and IWV estimates, related to both processing speed, in the processing and type of auxiliary observations to reach to IWV. There's also a small part discussing the relation the relation between IWV levels and patterns versus the occurence of specific cloud patterns in the region.

The quality of the presentation is high. In fact I found only one thing that really ought to be changed: There is a lot on information about how various approaches to the processing impacts ZTD and IWV quality, but reading the title potential readers will not appreciate that. The remedy is to either come up with a longer more precise title, or split the material up in two. In the latter case with a short description of the processing and the discussion with respect to clouds patterns, leaving the detailed accounting about quality versus processing for another manuscript.

We thank the referee for the kind comments and the positive assessment of the manuscript. We considered both options suggested by the referee: either coming up with a longer, more precise title, or splitting the material up in two. However, we think that both are inconvenient. We agree that the manuscript contains more than what the title reflects. But if we mention the comparison of different processing procedures, this would become too specific and leave out the fact that we also compare the GNSS estimates to other techniques (radiosondes, microwave radiometer) and ERA5, and that we discuss the relation of IWV with cloud patterns. A really comprehensive title in this case would be excessively long... The second option would not be satisfying either as the goal this paper is to describe the EUREC4A GNSS data as a whole. We think that in its present form the manuscript fits well our objectives and the scope of the journal. If keywords could be included this would help better reflect the content of the manuscript. However, the abstract properly fulfils this role as well.

SPECIFIC Early on tell that Grantley Adams airport is on Barbados.

Done line 11 and 43

Refer to BCON and BCOS already in section one, in order that all readers understand where Barbados is when you refer to fig 1.

Done line 58

Mention the proximity of BCO and the airport with the additional radiosone early on.

Done line 71-73

On page 6, move the sentence starting in line 153 up above line 150.

Done

Consider if you later uncertainty assessment requires the text in line 150-153, and if so, the text should be placed elsewhere.

The numerical values of partial derivates $\frac{\partial ZHD}{\partial Ps}$ and $\frac{\partial IWV}{\partial Ps}$ are used later in the paper when discussing the impact of Ps and Tm errors on ZHD and IWV (e.g. in line 198). However, we think it is more convenient to leave the derivatives equation in section 2.3 where the base equations are presented.

Line 199: Restrict the comment about ecmwf quality to the location. As you mention yourself later, ecmwf is grossly off wrt. topography at several of the other GNSS sites.

Done

Line 253: Would it be healthy to relax the constraint?

yes, this option may be considered in a future update. However, since this would change the characteristics of the ZTD data, and impact the subsequent data assimilation in NWP models, some testing is required beforehand. This point is beyond the scope of the paper, so we don't discuss it here.

page 17 and further. Several places you mention subpages of page 13 and 14, but they are not indicated on the figures. It should be OK just to refer to fig 13 and 14, otherwise introduce the a, b, c.. division somehow.

We added labels a, b, c... on the images

Section 5 appears to be partly discussion and partly conclusion.

the title has been changed to Discussion and conclusions.

Line 618 Antenna phase center models can result in a global bias I think. At least we saw a change in biases between GNSS and meteorological ZTDs when a significant change was made some years back.

Yes. This was already mentioned in the manuscript on line 621.

fig 9 Consider adding the location names

Done

fig 14 Enlarge the GNSS circles to make the figure easier to read.

Done

Table 2: It would be good to introduce a blank row before the IWV part of the table.

Done

**Reviewer 2**

This paper gives a very detailed description of GNSS measurements from 49 stations that were obtained during the EUREC4A campaign in the Caribbean Arc. The authors produced a 5 minute-temporal resolution dataset for each station for the purpose of scientific studies. The data for each station are available online in properly formatted user friendly NetCDF files. The authors compare this data to 4 operational streams of GNSS measurements, as well as to collocated measurements from a microwave radiometer and atmospheric soundings (two different types of sondes), as well as to ERA5. To demonstrate an application of the dataset, they link the profiles of integrated water vapor to different cloud patterns that occurred during EUREC4A. The description of all data and the data inter-comparison is done with great care. The authors report very interesting biases between the different data sets and provide hypotheses that may help explain the biases, even though no final answer is given, which is fine. This work should be highly useful to users, especially as the authors make specific suggestions which of the five GNSS streams might be most suitable for certain applications. In only have very minor comments.

We thank the referee for the kind comments and the positive feedback on of our work and the way it is presented in the manuscript.

Minor comments:

The authors suggest that biased GAIA observations caused a bias in ERA5 – a very interesting result. Is it possible to find out if they really were assimilated? BCO soundings were also assimilated in various operational analyses, but they do not have a dry bias. Given that BCO soundings occurred more frequently than GAIA soundings, that suggests ERA5 assimilated only GAIA, but not BCO? It would be really nice to know more about this and therefore I would like to encourage the authors to track the path of the data through the assimilation system of ERA5.

We further investigated this point with ECMWF staff (Irina Sandu, Alessandro Savazzi, and Mohamed Dahoui). The status is that, during EUREC$^4$A, ERA5 did not assimilate Vaisala RS41 soundings from the BCO but only the GAIA soundings (WMO code 78954) over Barbados, as well as most of the Vaisala RS41 soundings launched from the research vessels and the Vaisala RD41 dropsondes released from the research aircraft. On the other hand, the operational model (ECMWF IFS) assimilated all of the radiosondes and the dropsondes. The observation statistics for both ERA5 and IFS confirm that the GAIA soundings have a dry bias compared to the short-term forecast whereas the RS41 and RD41 sondes have a wet bias compared to the model. The latter result can be alternatively stated as the model having a dry bias compared to the RS41/RD41 sondes.

We added the following two sentences in the Discussion/conclusions section:

Line 626: "Indeed, ERA5 assimilated the GAIA sondes but not the sondes from the BCO during EUREC$^4$A (Irina Sandu, ECMWF, personal communication)".

Line 633: "Although the assimilation of biased radiosonde data might be thought as a potential reason, recent experiments with the ECMWF Integrated Forecasting System show that removing all the radiosondes and dropsondes in the EUREC$^4$A domain does not significantly impact the simulated humidity field (Savazzi and Sandu, personal communication)."

Already in the abstract, Flowers, Fish and Gravel are mentioned. I doubt that all readers are familiar with what they are. Perhaps keep it more general here and just speak of cloud organization.

We agree. This paragraph was changed to: "We classified the cloud organisation for five representative GNSS stations across the Caribbean Arc using visible satellite images. A statistically significant link was found between the cloud patterns and the local IWV observations from the GNSS sites as well as the larger-scale IWV patterns from the ECMWF reanalysis ERA5."

In the data there is geoid height and ellipsoid height. Could the authors please define the meaning in the text?

Geoid height is the conventional terminology for altitude or elevation, which is often referred to as mean sea level height (http://wiki.gis.com/wiki/index.php/Geoid). We changed the caption of Table S1 in Supplement, i.e. replacing 'altitude' by 'geoid height' to use the correct terminology and to be consistent with the content of the netcdf data files. We hope this clarifies the point.

(Figure) A4 in https://essd.copernicus.org/articles/13/491/2021/ shows that the air over sea is not more moist than that over Barbados. This is in contrast to the hypothesis at line 355.

Thank you for pointing this out. We changed the discussion accordingly (see lines 353 to 360 in the revised manuscript).

Figures often use a combination of green and red. Please ensure that this is color-blind friendly.

We checked all images for dichromatic vision (i.e. red, blue, or green blindness) and

they were still correctly legible. For your information, We used the web site below: https://www.color-blindness.com/coblis-color-blindness-simulator/

Typos:

Line 82: were $->$ where

Done

Line 633: biases $->$ biased

Done

Line 666: spatio-spatial ?

Done

general good $->$ general in good

Done